# PANDORA: DIFFUSION-BASED PROTEIN CONFORMATION GENERATION

## ABSTRACT

The field of protein design has garnered significant attention in AI for Science (AI4Science). While most existing studies focus on generating native protein structures for applications such as binding target discovery and drug design, this paper tackles a different problem. We aim to generate both native and non-native protein conformations, i.e., 3D structures along the folding pathway. This task holds significant potential for advancing key applications, including folding pathway prediction, drug discovery, and the study of protein misfolding diseases. To address this challenge, we introduce Pandora (i.e., Protein Conformation Diffusion Model for Generation), a novel diffusion-based framework designed to generate diverse, physically and chemically plausible protein backbone structures. By leveraging a diffusion architecture, Pandora captures a broad spectrum of folding patterns while adhering to biophysical constraints. Extensive experiments across multiple protein folding pathway datasets demonstrate the effectiveness and generalizability of our approach in producing realistic and biologically meaningful conformations. The implementation of Pandora is publicly available at https://anonymous.4open.science/r/Pandora-71AD.

## 1 INTRODUCTION

Proteins are fundamental biological macromolecules that govern a vast array of processes essential to life. Comprised of amino acid chains that fold into intricate three-dimensional structures, proteins exhibit diverse functionalities that are closely tied to their conformations. Understanding the geometric and structural properties of proteins is crucial for deciphering their mechanisms of action and functional diversity. Such insights are pivotal for driving innovations in areas like medicine (Kuhlman & Bradley, 2019), protein folding pathway (Majewski et al., 2023), and understanding protein misfolding disease (Hartl, 2017).

### 1.1 MOTIVATION

In recent years, breakthroughs like AlphaFold (Senior et al., 2020), AlphaFold 2 (Jumper et al., 2021), and the more recent AlphaFold 3 (Abramson et al., 2024) have significantly advanced our understanding of native protein structures. Beyond structural prediction, these landmark studies have catalyzed widespread interest in applying deep learning to biological sciences, sparking a surge of innovative research. For instance, methods such as RoseTTAFold (Baek et al., 2021), ESMFold (Lin et al., 2022), ProteinBERT (Brandes et al., 2022) showcase the transformative potential of deep learning in addressing complex biological problems, further establishing AI4Science as a driving force in protein research. While these advances have primarily targeted the prediction of static, native protein structures, they often overlook the dynamic folding pathways and transient intermediate conformations that proteins undergo during folding. Recent studies (Zheng et al., 2024; Wu et al., 2024; Ruzmetov et al., 2025) have investigated near-equilibrium conformations of folded proteins, offering important insights into the structural fluctuations of stable proteins. However, this line of research remains limited, as it fails to model the full folding process—particularly the transient, non-native conformations essential for understanding protein folding mechanisms.

Understanding these non-native conformations is crucial for uncovering the mechanisms behind protein folding and misfolding, which are directly linked to various diseases such as Alzheimer's and Parkinson's (Selkoe, 2004). The inability to fully capture the spectrum of protein conformations

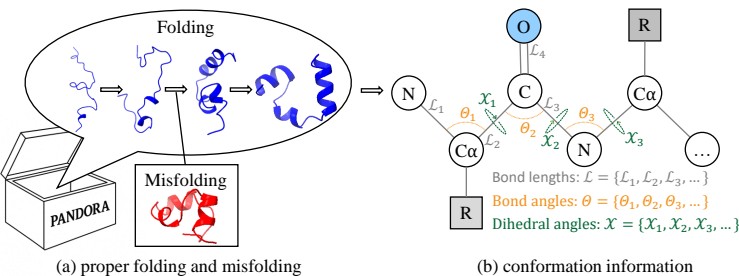

Figure 1: Different protein conformations.

highlights the need for novel computational frameworks capable of exploring both native and non-native protein structures. Traditional approaches to studying non-native conformations in the folding process primarily rely on molecular dynamics (MD) simulations (Binder et al., 2004), which solve differential equations using mathematical and physical models. However, such simulations are computationally expensive and time-prohibitive, particularly for larger proteins. For example, simulating a protein's folding process over $10\mu s$ can take months (Shaw et al., 2010), or even longer, due to the immense computational cost of resolving molecular interactions at extremely fine time steps. Addressing these limitations, the emergence of deep learning-based approaches offers new hope, enabling the modeling of longer timescales and the dynamics of larger proteins, thereby facilitating advancements across multiple domains.

*Protein Folding Mechanisms.* Understanding how proteins fold into their native structures (as shown in Figure 1 (a)) is essential for uncovering the principles that govern their functionality. Some research (e.g., (Zhao et al., 2023; Janson et al., 2023)) explore folding pathway by utilizing deep learning techniques, while (Majewski et al., 2023) accelerates dynamics via coarse-grained simulation. Recently, (Ianeselli et al., 2025) makes use of Variational autoencoders to predict folding actions. Understanding the process by which a protein transitions between conformations is essential, making the study of non-native conformations particularly important.

*Protein Misfolding Mechanisms.* Investigating protein misfolding mechanisms (as shown in Figure 1 (a)) is crucial for understanding the causes of various diseases, such as Alzheimer's, Parkinson's, and Huntington's disease. Misfolding occurs when proteins deviate from their intended folding pathways, resulting in the adoption of aberrant conformations that often aggregate into toxic structures. Some research (e.g., (Englander et al., 2007)) studies misfolding protein in the manner of experimental observations, while (Bhowmik et al., 2018) utilizes clustering methods to find potentially misfolded states among various non-native conformations. By leveraging deep learning methods that can sample a diverse ensemble of non-native conformations, we can better uncover elusive misfolded states, thereby enhancing our understanding of misfolding pathways and their role in disease mechanisms.

As highlighted above, advancing our understanding of protein folding and misfolding mechanisms requires the ability to generate a diverse range of both native and non-native conformations that reflect the intricacies of folding pathways. Producing a wide variety of native and non-native states provides several benefits. Firstly, it enables the discovery of critical intermediates that contribute to protein aggregation and misfolding-related diseases. Additionally, studying non-native conformations provides structural insights into the folding process, revealing how proteins transition between different states along their folding pathways. Therefore, *developing innovative methods to generate and analyze diverse non-native conformations is of paramount importance.*

## 1.2 CHALLENGES AND CONTRIBUTIONS

Native protein structure prediction has been well studied in many papers (e.g., (Senior et al., 2020; Jumper et al., 2021; Baek et al., 2021; Lin et al., 2022; Abramson et al., 2024)), and meanwhile some studies (e.g., (Zheng et al., 2024; Wu et al., 2024; Ruzmetov et al., 2025)) focus on states near the properly folded protein. In addition, some works have explored folding pathways, such as (Zhao et al., 2023; Janson et al., 2023; Majewski et al., 2023; Ianeselli et al., 2025). However, most of these works may not be capable of generating various non-native protein conformations. As a consequence, several challenges remain underexplored.

- *Challenge 1: Stochastic conformations.* Protein folding is an inherently complex process influenced by environmental factors, such as temperature, which introduces a degree of randomness into the folding pathways. Even when using MD simulations, the outcomes can vary and exhibit

stochasticity. Therefore, it is essential to develop models that inherently incorporate this randomness to better capture the diversity of folding behaviors. If we directly apply previous methods, such as AlphaFold 2 (Jumper et al., 2021), ESMFold (Lin et al., 2022), and so on, we can only obtain finite and determined protein conformations, which is not desirable to address this challenge. Some of other methods (e.g., (Janson et al., 2023; Ianeselli et al., 2025)) focusing on non-native conformations also fall into such inability. Thus, it is crucial to take stochasticity into account so as to generate diverse non-native conformations.

- *Challenge 2: Structurally valid conformations.* Ensuring that the generated conformations are not only diverse but also structurally valid is a significant challenge. Non-native conformations must satisfy physical and chemical constraints, such as steric clashes, bond angles, and dihedral angles, to remain biologically meaningful. Simply generating diverse conformations without considering structural validity may result in unrealistic, unstable, or biologically irrelevant structures. Some methods (e.g., (Huang et al., 2024)) generate molecules that may not be able to capture the special backbone structure of conformations, while some methods (e.g., (Zheng et al., 2024; Ruzmetov et al., 2025)) only concentrate on native proteins and may not be capable of modeling the diversity of non-native conformations. Furthermore, several methods (e.g., Lu et al. (2023); Wang et al. (2024); Shen et al. (2025)) generate conformations at the amino acid residue level, implicitly fixing certain bond lengths and bond angles during generation. However, prior studies (Lundgren & Niemi, 2012; Peng et al., 2014; Karplus et al., 2008) have shown that, for internal residues, key bond angles, particularly the $N - C\alpha - C$ angle, are not invariant but vary systematically with secondary structure. This suggests that treating such angles as rigid may overlook important conformation-dependent stereochemistry. Thus, it is essential to integrate mechanisms or constraints that can guarantee the structural plausibility of the generated conformations at the atomic level under the premise of diversity, with particular emphasis on maintaining atomic-level stereochemical fidelity.

**Our approach and contributions.** We propose Pandora (shorted for Protein Conformation Diffusion Model for Generation), a novel framework designed to generate diverse and structurally valid protein conformations. Pandora operates through two primary phases: the forward process and the reverse process. In the forward process, a conditional transformer model that fully embeds structural information is trained to denoise perturbed conformations. In the reverse process, a random conformation is progressively refined step by step using regulation techniques, ultimately producing a wide range of valid conformations. The key contributions of this paper are summarized as follows.

- **Non-native Protein conformation Generation.** To the best of our knowledge, this is the first work that explores generating diverse non-native protein conformations in protein folding pathways in atomic level. Generating such conformations is crucial for understanding intermediate folding states and misfolding mechanisms, as well as for advancing research in protein structure prediction and related applications.

- **An innovative method driven by the diffusion framework.** We develop an effective method, Pandora, to generate diverse conformations for different proteins. For one thing, Pandora considers intrinsic connection information (i.e., amino acids, atoms, chemical bonds) since this information is identical for the same protein. For another, Pandora also incorporates different structural information (i.e., bond lengths, bond angles, and dihedral angles), as these geometric differences are fundamental in determining the diverse conformations of an identical protein.

- **Experimental results.** We perform comprehensive experiments on multiple protein folding pathway datasets to assess the generation capability of Pandora. Experimental results show that our algorithm outperforms baseline methods by a significant margin while maintaining robust generalizability.

## 2 PROBLEM DEFINITION AND OVERVIEW

We first define the protein conformation generation problem and then present the overview of our proposed Pandora. Appendix A.1 lists the frequently used notations and terminologies in the paper.

### 2.1 PROBLEM DEFINITION

**Protein.** Proteins consist of chains of amino acid residues. Each residue contains an alpha carbon ($C\alpha$) atom, which is bonded to an amino group ($-NH_2$), a carboxyl group ($-COOH$), and a side-chain ($-R$) that determines the residue type. Consecutive residues are linked by peptide

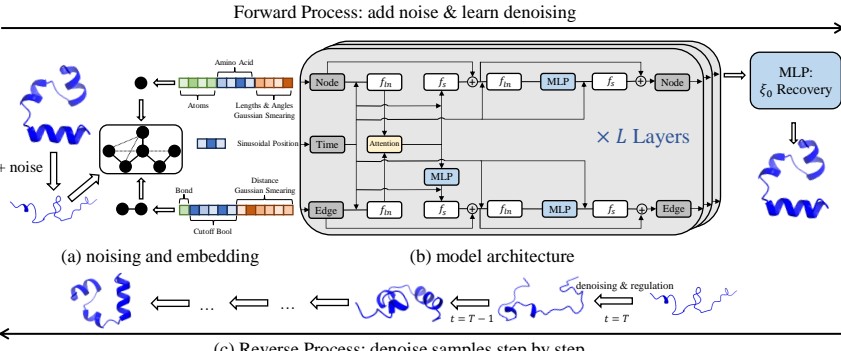

(a) noising and embedding      (b) model architecture

(c) Reverse Process: denoise samples step by step

Figure 2: Overview of Pandora.

bonds, formed through a dehydration synthesis process. A protein sequence can be represented as $\mathcal{S} = [s_1, s_2, ..., s_n]$, where n is the number of residues, and $s_i \in \{0, ..., 19\}$ indicates the type of the $i$-th residue.

**Backbone.** The protein backbone is composed of the nitrogen ($N$) from the amino group, the alpha carbon ($C\alpha$), the carbonyl carbon ($C$), and the carbonyl oxygen ($O$). The backbone structure is the most essential part of a protein. Given the backbone, the full protein conformation can be reconstructed using various tools (e.g., (Krivov et al., 2009; Zhang et al., 2023)).

**Conformation.** A protein's conformation refers to its 3D structure, which can be represented as a series of atomic coordinates defining the spatial arrangement of its atoms. Then, conformation can be formulated as $COOR = [coor_1, coor_2, ..., coor_n]$, where $coor_i$ denotes the set of atom coordinates belonging to the $i$-th residue, while the backbone coordinates are thereby $COOR^{(bb)}$.

To generate biologically plausible conformations, the bond lengths $\mathcal{L}$ (as shown in Figure 1 (b)) between atoms must adhere to their natural distributions. Otherwise, deviations in bond lengths can lead to an increase in bond energy, resulting in unrealistic conformations. Similarly, bond angles $\Theta$ and dihedral torsions (i.e., dihedral angles $\mathcal{X}$) (as shown in Figure 1 (b)) play a crucial role in determining how a protein folds and twists. Therefore, the bond angles and dihedral angles of generated conformations must also conform to their natural distributions. Detailed definitions of $\mathcal{L}$, $\Theta$, and $\mathcal{X}$ can be found in Appendix A.1 .

Note that given the atomic coordinates $COOR^{(bb)}$, one can derive $(\mathcal{L}, \Theta, \mathcal{X})$, and conversely, given $(\mathcal{L}, \Theta, \mathcal{X})$, the corresponding coordinates can also be reconstructed, though additional rotation and translation may be required. In summary, $COOR^{(bb)} \Leftrightarrow (\mathcal{L}, \Theta, \mathcal{X})$ under the assumption of rotation and translation invariance. Since protein conformations remain unchanged under any rotation or translation, generating $(\mathcal{L}, \Theta, \mathcal{X})$ directly avoids the invariance problem that arises when generating coordinates directly. Based on this, the problem can be formally defined as follows.

**Problem Statement.** Given the types of amino acids $\mathcal{S}$ of a protein conformation, the task is to find a model $\Phi$ that can help generate $\xi \triangleq (\mathcal{L}, \Theta, \mathcal{X})$ that conform to biological plausibility. That is, we aim to model the conditional distribution $p_\Phi(\xi|\mathcal{S}) \triangleq p_\Phi(\mathcal{L}, \Theta, \mathcal{X}|\mathcal{S})$.

## 2.2 OVERVIEW OF PANDORA

We present the overview of Pandora in Figure 2. There are two main phases in Pandora.

**Forward Process.** We first train a denoising model that can denoise the noised conformations, which consists of two main steps.

Noising and Embedding. To fully utilize conformation information, we initially construct a graph for the noised conformations, and then embed the graph with amino acid types, bond information, and angle information in nodes and edges as shown in Figure 2 (a).

Conditional Transformer. As depicted in Figure 2 (b), we employ conditional transformer blocks to update the node and edge features by mutual deep interaction. Meanwhile, conditional information is also incorporated to enhance the representation of nodes and edges. Based on the acquired representation, we calculate the denoised bond lengths, bond angles, and dihedral angles by simple Multi-Layer Perceptron (MLP) layers (Rosenblatt, 1958).

**Reverse Process.** Given a random structure as presented in Figure 2 (c), we apply the denoising model to denoise random conformations steps by steps. During each step, we regulate the conformation to ensure plausibility and add noise to introduce stochasticity.

## 3 METHODOLOGY

In this section, we present Pandora, Protein Conformation Diffusion Model for Generation). We first give the formulation of diffusion process on conformation graphs in Sec. 3.1 and then provide the details of Pandora architecture in Sec. 3.2. Finally, we introduce the inference procedure in Sec. 3.3.

### 3.1 DIFFUSION MODELS

Denoising diffusion probabilistic models (Ho et al., 2020) are a type of generative model that captures the underlying data distribution of $\xi$ (defined in Sec. 2.1) by employing a forward process and a reverse process. Following (Song et al., 2020; Jing et al., 2022), we aim to construct a diffusion process $\{\xi_t\}_{t=0}^T$ indexed by a continuous time variable $t \in [0, T]$, such that $\xi_0 \sim p_0$ (i.e., data distribution), for which we have a dataset of independent and identically distributed (i.i.d.) samples. Then, This diffusion process can be modeled as the solution to an Itô Stochastic Differential Equation (SDE), given by the following formula:

$$d\xi = \mathbf{f}(\xi, t)dt + g(t)d\mathbf{w}. \tag{1}$$

where $w$ is the Wiener process, $\mathbf{f}(\xi, t) : \mathbb{R}^{|\xi|} \to \mathbb{R}^{|\xi|}$ is the drift coefficient of $\xi_t$ and $g(t)$ is the diffusion coefficient of $\xi_t$. Given $\xi_T \sim p_T$ (i.e., prior distribution), for which we have a tractable form to generate samples efficiently, the reverse process samples from the prior and yields samples from the data distribution $p_0(\xi)$ by solving the reverse SDE.

$$d\xi = \left[\mathbf{f}(\xi_t, t) - g^2(t)\nabla_\xi \log p_t(\xi)\right]dt + g(t)d\bar{\mathbf{w}}. \tag{2}$$

Here, $\bar{\mathbf{w}}$ is a standard Wiener process when time flows backwards from $T$ to 0, and $dt$ is an infinitesimal negative timestep. With definition above, our goal is to train a neural network $\Phi$ to fit the score $\nabla_\xi \log p_t(\xi)$. Inspired by the techniques applied in (Huang et al., 2024), we train $\Phi$ to recover $\xi_0$ by optimizing the following objective function.

$$\min_\theta \mathbb{E}_t \left\{ \lambda(t)\mathbb{E}_{\xi_0}\mathbb{E}_{\xi_t|\xi_0}\left[||\Phi_\theta(\xi_t, t) - \xi_0||_2^2\right]\right\} \tag{3}$$

With proper choice of $\lambda(t)$ (Huang et al., 2024; Song et al., 2020), it is equivalent to the simple noise prediction loss used in (Ho et al., 2020).

### 3.2 NETWORK ARCHITECTURE

There are two main stages, including Noising and Embedding (in Sec. 3.2.1) and Conditional Transformer (in Sec. 3.2.2).

#### 3.2.1 NOISING AND EMBEDDING

**Noising.** For a given backbone of a protein conformation, we first obtain its bond length, bond angle, and dihedral angle information, marked by $\xi_0 = (\mathcal{L}_0, \Theta_0, \mathcal{X}_0)$. Then, we add noise with different levels corresponding to $t$ by the following formula:

$$\xi_t = f_{\text{cut}}\left(\sqrt{\bar{\alpha}_t} \cdot \xi_0 + \sqrt{1 - \bar{\alpha}_t} \cdot \epsilon\right) \tag{4}$$

where $\epsilon$ is the Gaussian noise, while definition of $\bar{\alpha}_t$ and $\alpha/\beta$ schedule are detailed in Appendix A.2. Note that a bond angle only ranges in $[0, \pi]$, while a dihedral angle only ranges in $[-\pi, \pi)$, which means that we must regulate $\Theta_t$ and $\mathcal{X}_t$ to maintain consistency. For simplicity, we directly apply a cutoff operation $f_{\text{cut}}$ so that $\Theta_t$ and $\mathcal{X}_t$ remain within their respective valid ranges.

**Embedding.** To fully utilize information, we embed both node (i.e., atom) and edge (i.e., a pair of nodes) information by processing $x_i$ (i.e., all information of the $i$-th atom).

*Node Embedding.* For each atom, we embed atom types, amino acid types, and $\xi_t$ into a high-dimensional representation. Atom types and amino acid types are mapped into high-dimensional space using the embedding functions $f_{at}^e : \{C, C\alpha, N, O\} \to \mathbb{R}^{l_{at}}$ and $f_{\mathcal{S}}^e : \mathcal{S} \to \mathbb{R}^{l_s}$, respectively. Meanwhile, $\xi_t = (L_t, \Theta_t, \mathcal{X}_t)$ is transformed using Gaussian smearing techniques (Unke & Meuwly,

2019), which smooth discrete data (e.g., bond length, bond angle, dihedral angle) into continuous distributions. This involves applying several Exponential Gaussian Kernels given by:

$$f_{\mathcal{L}}^e, f_{\Theta}^e, f_{\mathcal{X}}^e(x) \triangleq \left\{ \exp\left(\beta_k^e \cdot (x - \mu_k^e)^2\right) \right\}_{k=1}^K : \mathbb{R} \to \mathbb{R}^K \tag{5}$$

where the definitions of $\beta_k^e$, $\mu_k^e$, and $K$, as well as additional details about the embedding functions, are provided in Appendix A.3.. Consequently, the embedding of node $i$ is obtained by concatenating the above representations, i.e., $x_i^e \triangleq [f_{at}^e; f_{\mathcal{S}}^e; f_{\mathcal{L}}^e; f_{\Theta}^e; f_{\mathcal{X}}^e](x_i) \in \mathbb{R}^{l_{ne}}$

*Edge Embedding.* For each edge between a pair of nodes, we also construct effective edge embeddings by mapping them to high-dimensional representations. First, we check whether there is a chemical bond by $f_{cb}^e \in \{0, 1\}$. Then, we compute the distance $d_{ij} = f_d^e(x_i, x_j)$ between atom $i$ and $j$ via a distance function. Similarly, we leverage Gaussian smearing $f_{dr}^e : \mathbb{R} \to \mathbb{R}^K$ to smooth the distance relation. Meanwhile, we further discretize distance to capture additional relationships between atom pairs by a series of boolean functions $f_b^e(d_{ij}) \triangleq \{I(d_{ij} \leq d_k)\}_{k=1}^K$, where $d_k$'s are a series of thresholds illustrated in Appendix A.3. In summary, the embedding of edge is obtained by concatenating the above terms, i.e., $x_{ij}^e \triangleq [f_{cb}^e; f_d^e; f_{dr}^e; f_b^e](x_i, x_j) \in \mathbb{R}^{l_{ee}}$

### 3.2.2 CONDITIONAL TRANSFORMER

We first introduce the core attention mechanism, then illustrate how time information is embedded into the conditional architecture, and finally explain how to recover $\xi_0$.

**Attention Mechanism.** Inspired by (Huang et al., 2024), we also extend Transformer (Vaswani et al., 2017) with the assistance of edge information. The procedure in the *l-th* layer is as follows:

$$q_i^{(l)}, k_i^{(l)}, v_i^{(l)} = \text{FC}_{q,k,v}^{(l)}(x_i^{(l)}), \quad q_{ij}^{(l)}, v_{ij}^{(l)} = \text{FC}_{q,v}^{(l)}(x_{ij}^{(l)})$$

$$w_{ij}^{(l)} = (\tanh(q_{ij}^{(l)}) \odot q_i^{(l)}) \cdot k_j^{(l)}, \quad m_i^{(l)} = \sum_j w_{ij}^{(l)}(\tanh(v_{ij}^{(l)}) \odot v_i^{(l)}) \tag{6}$$

where FC is a fully connected layer. The above process is denoted as $Att(x_i^{(l)}, x_{ij}^{(l)})$.

**Conditional Architecture.** Directly concatenating time information into node or edge embedding may fail to let them fully interact. To address this, we implement two useful functions proposed in (Huang et al., 2024), i.e., $f_{ln}(x, c) = (1 + \text{MLP}(c)) \cdot \text{LN}(x) + \text{MLP}(c)$ and $f_s(x, c) = \text{MLP}(c) \cdot h$. Here, LN refers to layer normalization (Ba et al., 2016) and $c$ is the conditional information of $t$, which is embedded using Sinusoidal Position Embeddings (Vaswani et al., 2017). Then, we can define multihead attention blocks with conditional $m_i^{(l)} = Att(f_{ln}(x_i^{(l)}, c), f_s(x_{ij}^{(l)}, c))$. Initially, $x_i^{(0)}$ and $x_{ij}^{(0)}$ are set to $x_i^e$ and $x_{ij}^e$, respectively. By Equation (6), we update node and edge information by the following equations:

$$\widehat{x}_i^{(l)} = f_s(m_i^{(l)}, c) + x_i^{(l)}, \quad x_i^{(l+1)} = f_s\left(\text{MLP}(f_{ln}(\widehat{x}_i^{(l)}), c), c\right) + \widehat{x}_i^{(l)}$$

$$\widehat{x}_{ij}^{(l)} = f_s\left(\text{FC}(m_i^{(l)} + m_j^{(l)}), c\right) + x_{ij}^{(l)}, \quad x_{ij}^{(l+1)} = f_s\left(\text{MLP}(f_{ln}(\widehat{x}_{ij}^{(l)}), c), c\right) + \widehat{x}_{ij}^{(l)} \tag{7}$$

$\xi_0$ **Recovery.** After $L$ layers of conditional attention layers, we obtain $x_i^{(L)}$ for node $i$. Inspired by (Zhang et al., 2022), for an arbitrary bond length $\xi_{ij}$, bond angle $\xi_{ijk}$, or dihedral angle $\xi_{ijkl}$, we compute its change $\Delta\xi_{ij...} \triangleq \text{MLP}(\text{mean}(x_i^{(L)}, x_j^{(L)}, ...))$. For instance, when calculating dihedral angles, the offset is computed as $\Delta\xi_{ijkl} = \text{MLP}((x_i^{(L)} + x_j^{(L)} + x_k^{(L)} + x_l^{(L)})/4)$. Finally, we can acquire $\widehat{\xi}_0$ by the following equations:

$$\widehat{\xi}_0 \triangleq \Phi_\theta(\xi_t, t) = f_{\text{cut}}(\xi_t + \Delta\xi) \tag{8}$$

### 3.3 INFERENCE

The inference process of Pandora is adapted from the sampling process in (Ho et al., 2020), given by the following equations:

$$\widehat{\epsilon} = \frac{\xi_t - \sqrt{\bar{\alpha}_t} \cdot \widehat{\xi}_0}{\sqrt{1 - \bar{\alpha}_t}}, \quad \tilde{\xi}_{t-1} = \frac{\xi_t - \frac{1 - \bar{\alpha}_t}{\sqrt{1 - \bar{\alpha}_t}}\widehat{\epsilon}}{\sqrt{\alpha_t}}, \quad \xi_{t-1} = f_{\text{cut}}(\tilde{\xi}_{t-1} + \sigma_t \epsilon \cdot I(t > 1)) \tag{9}$$

Table 1: Comparison on $\xi_0$ recovery and ablation study.

| Methods | JODO | JODO-L | GearNet | STR2STR | CONFD. | CONFR. | Pandora | - len | - angle | - torsion | - type |
|---------|------|--------|---------|---------|--------|--------|---------|-------|---------|-----------|--------|
| $\text{MAE}_{\mathcal{L}}$ ↓ | 0.176 | 0.093 | 0.422 | 0.526 | 0.337 | 0.157 | **0.025** | 0.168 | 0.026 | 0.025 | 0.025 |
| $\text{MAE}_{\Theta}$ ↓ | 0.293 | 0.218 | 0.430 | 0.375 | 0.252 | 0.201 | **0.050** | 0.052 | 0.190 | 0.051 | 0.051 |
| $\text{MAE}_{\mathcal{X}}$ ↓ | 1.415 | 1.369 | 0.753 | 1.167 | 1.083 | 0.702 | **0.374** | 0.380 | 0.382 | 0.688 | 0.446 |

As the time steps $t$ decrease from $T$ to 1, we can obtain $\xi_0$ at $t = 1$. Note that during each reverse step, we perform cutoff functions twice: once in Equation (8) and once in Equation (9), so that we can regulate the generated conformation to maintain consistency and ensure that the conformations remain coherent throughout the generation process.

## 4 EXPERIMENTS

### 4.1 EXPERIMENTAL SETUP

**Datasets.** In our experiments, we use molecular dynamics simulation datasets from (Lindorff-Larsen et al., 2011). We use five datasets for training (Villin, WW domain, Protein B, BBL, and Homeodomain) and two datasets for testing and generalization evaluation (Trp-cage and BBA), ensuring that the test proteins are unseen during training. Appendix B.1 provides a brief description of each dataset and reports the numbers of atoms, amino acids, and conformations. Details of hyperparameters are provided in Appendix B.2.

**Algorithms.** We evaluate the performance of several methods for molecular conformation generation. As a representative baseline, we include JODO (Huang et al., 2024), an atomic coordinate generation model. We also introduce an enhanced variant, JODO-L, which incorporates an additional $\xi_0$ loss term to improve conformational plausibility. For further comparison, we include several additional baselines: GearNet (as implemented in DiffPack (Zhang et al., 2022)), STR2STR (Lu et al., 2023), CONFDIFF (Wang et al., 2024), and CONFROVER (Shen et al., 2025). Further details on all baseline methods are provided in Appendix B.3.

**Evaluation Metrics.** We evaluate the quality of $\xi_0$ recovery using three metrics, including bond length mean absolute error ($\text{MAE}_{\mathcal{L}}$), bond angle MAE ($\text{MAE}_{\Theta}$), and dihedral angle MAE ($\text{MAE}_{\mathcal{X}}$). As for evaluation of inference, we compare the generated $\xi_0$ distribution with the true $\xi_0$ distribution by histogram, the Wasserstein distance (Villani, 2021), and probability density map.

We conduct experiments on a Linux server with an AMD EPYC 9654 96-Core CPU, 1TB of RAM, and 8 Nvidia RTX A6000 GPUs. We provide the codes The codes can be accessed at https://anonymous.4open.science/r/Pandora-71AD.

### 4.2 $\xi_0$ RECOVERY

Table 1 shows the performance comparison (averaged over three trials) of Pandora and the baselines on $\xi_0$ recovery across five datasets, while standard deviations and results for each dataset are provided in Appendix B.4. In summary, Pandora consistently outperforms the baseline methods in recovering the lengths and angles of different conformations with marginal fluctuation. This superior performance highlights Pandora's ability to better capture structural variations and accurately reconstruct conformations, making it a reliable tool for such tasks.

As for other methods, JODO-L outperforms JODO across all three metrics, primarily due to the incorporation of the $\xi_0$ loss in JODO-L. While this modification enhances its performance, the dihedral recovery still exhibits a significantly larger MAE compared to Pandora. On the other hand, GearNet achieves a relatively smaller $\text{MAE}_{\mathcal{X}}$ but suffers from higher MAE in bond length and bond angle recovery. Additionally, STR2STR delivers moderate results but incurs larger errors in $\text{MAE}_{\mathcal{L}}$ and $\text{MAE}_{\Theta}$. CONFDIFF (CONFD.) improves over STR2STR across all three metrics, with a notable gain on $\text{MAE}_{\Theta}$, yet still trails Pandora. CONFROVER (CONFR.) further reduces errors relative to CONFDIFF, achieving the strongest baseline performance on bond angle and dihedral recovery.

### 4.3 ABLATION STUDY

As described in Sec. 3.2.1, the node embedding consists of several components. To evaluate the impact of each component, we measure the performance by removing specific parts, including amino acid types (-type), bond length (-len), bond angle (-angle), dihedral angle (-torsion).

Table 2: Wasserstein distance between true and generated distribution on training dataset.

| Methods | $\mathcal{L}_1 \downarrow$ | $\mathcal{L}_2 \downarrow$ | $\mathcal{L}_3 \downarrow$ | $\mathcal{L}_4 \downarrow$ | $\Theta_1 \downarrow$ | $\Theta_2 \downarrow$ | $\Theta_3 \downarrow$ | $\Theta_4 \downarrow$ | $\mathcal{X}_1 \downarrow$ | $\mathcal{X}_2 \downarrow$ | $\mathcal{X}_3 \downarrow$ | $\mathcal{X}_4 \downarrow$ |
|---|---|---|---|---|---|---|---|---|---|---|---|---|
| JODO | 0.072 | 0.065 | 0.052 | 0.091 | 0.191 | 0.102 | 0.133 | 0.094 | 0.742 | 0.394 | 1.311 | 0.884 |
| JODO-L | 0.078 | 0.057 | 0.061 | 0.078 | 0.208 | 0.180 | 0.185 | 0.159 | 0.751 | 0.567 | 1.317 | 0.885 |
| GearNet | 1.438 | 1.425 | 1.444 | 1.438 | 1.097 | 1.006 | 0.905 | 0.951 | 2.878 | 2.543 | 1.091 | 2.977 |
| Pandora | **0.023** | **0.020** | **0.017** | **0.022** | **0.022** | **0.021** | **0.018** | **0.014** | **0.121** | **0.153** | **0.088** | **0.163** |

As shown in Table 1, removing bond length, bond angle, or dihedral angle Gaussian smearing leads to a significant increase in the corresponding MAE, while standard deviations are provided in Appendix B.4. For instance, if bond length Gaussian smearing is removed, $\text{MAE}_\mathcal{L}$ increases by approximately 6 times. Similarly, $\text{MAE}_\Theta$ and $\text{MAE}_\mathcal{X}$ increase by 3 times and 1 time, respectively, when the corresponding components are removed. However, the changes in the remaining MAE values are negligible. Regrading removal of amino acid types, both $\text{MAE}_\mathcal{L}$ and $\text{MAE}_\Theta$ change marginally, while $\text{MAE}_\mathcal{X}$ exhibits a significant increase. This is likely because different amino acid types influence the dihedral angles of peptide bonds formed during dehydration condensation. The information encoded in amino acid types plays a crucial role in guiding the prediction of dihedral angles. Once this information is removed, the model struggles to accurately predict these angles, leading to a noticeable decline in performance.

## 4.4 INFERENCE

The detailed definitions of $\mathcal{L}$, $\Theta$, and $\mathcal{X}$ can be found in Appendix A.1. Note that methods such as STR2STR, CONFDIFF, and CONFROVER operate at the residue level. As a result, certain bond lengths, bond angles, and dihedrals are fixed during generation, making distributional comparisons on these variables non-informative. We therefore report average Wasserstein distances across five datasets in Table 2. The complete results for all methods are provided are provided in Appendix B.5.

From the results above, we can conclude that Pandora consistently exhibits small Wasserstein distances across all metrics, indicating its ability to generate conformations with a realistic distribution that closely matches the ground truth. Additionally, although JODO-L outperforms JODO in $\xi_0$ recovery, the scenario is different here. Specifically, JODO achieves better performance than JODO-L in terms of $\Theta$ and $\mathcal{X}$. This is likely because the modifications in JODO-L, while effective in reducing $\text{MAE}\Theta$ and $\text{MAE}\mathcal{X}$, do not translate effectively to $\Theta$ and $\mathcal{X}$ inference. As for GearNet, it struggles to generate conformations that align with the true distribution, particularly in bond length ($\mathcal{L}$), where its performance is notably weaker.

To gain deeper insight into how Pandora effectively generates realistic conformations, we provide histograms of the true and generated distributions for $\mathcal{L}$, $\Theta$, and $\mathcal{X}$ for all datasets in Appendix B.5. Histograms for other datasets across all methods are also presented in Appendix B.5. Figure 5, analogous to Figure 3, demonstrates that Pandora's generated distributions closely align with true distributions, particularly for $\Theta$ and $\mathcal{X}$. While the $\mathcal{L}$ distributions exhibit a noticeable rightward shift, the magnitude of this shift is approximately +0.025Å, which aligns with $\text{MAE}_\mathcal{L}$ and is negligible when compared to typical bond lengths of 1.1Å to 1.7Å. Furthermore, Pandora effectively captures the asymmetry in $\mathcal{X}_1$, demonstrating its ability to recognize chirality in amino acids. In contrast, other methods, such as JODO, tend to produce symmetric dihedral angle distributions, thereby ignoring chirality, as illustrated in Figure 10 in Appendix B.5.

## 4.5 GENERALIZATION

We evaluate the generalizability of Pandora using two additional protein datasets that were excluded from the training set, meaning these proteins were never seen during training. As shown in Table 3, Pandora successfully generates plausible conformational structures for unseen proteins while maintaining a low Wasserstein distance, demonstrating its strong generalization capability. For a clearer illustration, histograms for dataset BBA are presented in Figure 3, while those for dataset Trp-cage are provided in Appendix B.6. As shown in the figures, the generated distributions closely align with the true distributions, further validating Pandora's generalizability. Moreover, we evaluate joint distributions over the radius of gyration (Rg, a compactness measure) and the root mean square deviation (RMSD, a deviation from a reference after alignment). As shown in Figure 4, Pandora's samples closely match the true Rg–RMSD landscape, reproducing the high-density ridge, its orientation, and the spread toward higher RMSD, further supporting Pandora's generalizability. Probability density maps for other datasets are provided in Appendix B.7.

Table 3: Wasserstein distance between true and generated distribution on inference set.

| Datasets | $\mathcal{L}_1 \downarrow$ | $\mathcal{L}_2 \downarrow$ | $\mathcal{L}_3 \downarrow$ | $\mathcal{L}_4 \downarrow$ | $\Theta_1 \downarrow$ | $\Theta_2 \downarrow$ | $\Theta_3 \downarrow$ | $\Theta_4 \downarrow$ | $\mathcal{X}_1 \downarrow$ | $\mathcal{X}_2 \downarrow$ | $\mathcal{X}_3 \downarrow$ | $\mathcal{X}_4 \downarrow$ |
|---|---|---|---|---|---|---|---|---|---|---|---|---|
| BBA | 0.024 | 0.021 | 0.018 | 0.021 | 0.025 | 0.026 | 0.014 | 0.011 | 0.585 | 0.533 | 0.183 | 0.673 |
| Trp-cage | 0.023 | 0.019 | 0.017 | 0.021 | 0.024 | 0.018 | 0.020 | 0.014 | 0.268 | 0.134 | 0.105 | 0.262 |

Figure 3: Pandora-generated histograms for dataset BBA.

Figure 4: Pandora-generated probability density map for dataset Trp-cage.

## 5 RELATED WORK

**Diffusion Models for Molecule Generation.** Diffusion models find extensive applications across diverse areas of molecule generation. There are many works focusing on molecule generation, such as (Hoogeboom et al., 2022; Jing et al., 2022; Wu et al., 2022; Huang et al., 2023; Xu et al., 2022; Wang et al., 2025), demonstrating that diffusion models have been successfully applied to molecular design, yielding remarkable results. Proteins, as a unique class of molecules, have also garnered significant attention in the field of protein design. Thus, many works (Anand & Achim, 2022; Wu et al., 2024; Ingraham et al., 2023; Watson et al., 2023; Lin & AlQuraishi, 2023; Luo et al., 2022; Watson et al., 2022; Yim et al., 2023; Lu et al., 2023; Wang et al., 2024; Shen et al., 2025) apply diffusion models to tackle diverse tasks. These advancements highlight the growing impact of diffusion models in advancing molecular and protein science.

**Protein Folding and Misfolding.** Protein folding remains a fundamental challenge in molecular biology that has yet to be fully resolved. Correctly folded proteins are essential for nearly all biological functions, and failures in folding can lead to severe consequences, such as loss of function or the aggregation of misfolded proteins. Numerous studies have explored folding pathways through both theoretical and experimental approaches (Onuchic & Wolynes, 2004; Dill et al., 2008; Lindorff-Larsen et al., 2011), while other works focus on generating trajectories using deep learning methods Shen et al. (2025); Jing et al. (2024). Meanwhile, significant attention has been given to understanding the mechanisms of protein misfolding (Dobson, 2003; Englander et al., 2007; Bhowmik et al., 2018; Hartl, 2017), which is often implicated in neurodegenerative diseases like Alzheimer's, Parkinson's, and Huntington's disease (Selkoe, 2004). These efforts collectively aim to deepen our understanding of protein behavior and pave the way for advancements in addressing misfolding-related diseases.

## 6 CONCLUSION

This paper focuses on the generation of backbone structures for both native and non-native protein conformations. To generate diverse and biologically plausible protein conformations, we propose Pandora, a versatile framework that fully integrates comprehensive information and leverages a conditional transformer architecture. Pandora employs proper regulation during inference, enabling the production of desirable conformations. Extensive experiments demonstrate the effectiveness of Pandora in generating diverse protein conformations and validate the practicality and generalizability of the proposed model.

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

# APPENDIX

# A ADDITIONAL DETAILS IN METHODS

## A.1 NOTATIONS AND TERMINOLOGIES

Table 4 lists the frequently used notations and terminologies in this paper.

## A.2 $\alpha/\beta$ SCHEDULE

In this subsection, we introduce the $\alpha/\beta$ schedule set in Pandora. Following (Nichol & Dhariwal, 2021), we use the following setting.

$$f(t) = \cos^2\left(\frac{t/T + s}{1 + s} \cdot \frac{\pi}{2}\right), \quad \bar{\alpha}_t = \frac{f(t)}{f(0)}, \quad \beta_t = 1 - \alpha_t = 1 - \frac{\bar{\alpha}_t}{\bar{\alpha}_{t-1}} \tag{10}$$

where $s = 0.008$ is a small offset to prevent $\beta_t$ from being too small near $t = 0$, and $\beta_t$ is clipped to $[0.0001, 0.9999]$. As for $\sigma_t$ in Equation (9), we apply posterior variance $\tilde{\beta}_t$ given by the following formula.

$$\sigma_t = \sqrt{\hat{\beta}_t}, \quad \hat{\beta}_t = \frac{(1 - \bar{\alpha}_{t-1})}{1 - \bar{\alpha}_t} \cdot \beta_t \tag{11}$$

## A.3 EMBEDDING SETTINGS

As for exponential Gaussian kernels in Equation (5), we first define a set of $K$ values $\{\mu_k^e\}_{k=1}^K$ that are equally spaced between a minimum value $\mu_{\min}$ and a maximum value $\mu_{\max}$, as well as a corresponding coefficient $\beta_k^e$ for each interval:

$$\mu_k^e = \mu_{\min} + \frac{\mu_{\max} - \mu_{\min}}{K - 1} \cdot (k - 1), \quad k = 1, 2, \ldots, K,$$
$$\beta_k^e = -\frac{1}{2}\left(\frac{\mu_{\max} - \mu_{\min}}{K - 1}\right)^{-2}, \quad k = 1, 2, \ldots, K, \tag{12}$$

For $f_{\mathcal{L}}^e$, $f_{\Theta}^e$, $f_{\mathcal{X}}^e$, and $f_{\mathrm{dr}}^e$, we adopt different settings for $\mu_{\min}$, $\mu_{\max}$, and $K$. The specific configurations for each case are summarized in Table 5.

With respect to the boolean function $f_b^e$, we set the threshold values as follows:

$$d_k = \frac{5}{K - 1} \cdot (k - 1), \quad k = 1, 2, \ldots, K \tag{13}$$

where $K = 8$.

Table 4: Notations and terminologies

| Notations | Explanation |
|---|---|
| $\mathcal{L}_1$ | Bond length of $N_i - C\alpha_i$ |
| $\mathcal{L}_2$ | Bond length of $C\alpha_i - C_i$ |
| $\mathcal{L}_3$ | Bond length of $C_i - N_{i+1}$ |
| $\mathcal{L}_4$ | Bond length of $C_i - O_i$ |
| $\mathcal{L}$ | Union set of $\{\mathcal{L}_1, \mathcal{L}_2, \mathcal{L}_3, \mathcal{L}_4\}$ |
| $\Theta_1$ | Bond angle of $N_i - C\alpha_i - C_i$ |
| $\Theta_2$ | Bond angle of $C\alpha_i - C_i - N_{i+1}$ |
| $\Theta_3$ | Bond angle of $C_i - N_{i+1} - C\alpha_{i+1}$ |
| $\Theta_4$ | Bond angle of $C\alpha_i - C\alpha_i - O_i$ |
| $\Theta$ | Union of $\{\Theta_1, \Theta_2, \Theta_3, \Theta_4\}$ |
| $\mathcal{X}_1$ | Dihedral angle of $N_i - C\alpha_i - C_i - N_{i+1}$ |
| $\mathcal{X}_2$ | Dihedral angle of $C\alpha_i - C_i - N_{i+1} - C\alpha_{i+1}$ |
| $\mathcal{X}_3$ | Dihedral angle of $C_i - N_{i+1} - C\alpha_{i+1} - C_{i+1}$ |
| $\mathcal{X}_4$ | Dihedral angle of $N_i - C\alpha_i - C_i - O_i$ |
| $\mathcal{X}$ | Union of $\{\mathcal{X}_1, \mathcal{X}_2, \mathcal{X}_3, \mathcal{X}_4\}$ |
| $\xi$ | Union of $\{\mathcal{L}, \Theta, \mathcal{X}\}$ |
| $s_i$ | The type of an amino acid residue. |
| $\mathcal{S}$ | Union set of $\{s_i\}$, i.e., protein sequence. |
| $coor_i$ | The set of atom coordinates belonging to the $i$-th residue. |
| $COOR$ | Union of $\{coor_i\}$, i.e., coordinates of all atoms. |
| $COOR^{(bb)}$ | The backbone coordinates. |
| $\Phi, p_\Phi$ | A denoising,generative model. |
| $T$ | Total number of time steps. |
| $t$ | Time index, $t = 0, 1, \ldots, T-1$. |
| $p_0, p_T$ | The data distribution and the prior distribution, respectively. |
| $\xi_0, \xi_T$ | A sample drawn from $p_0$ and $p_T$. |
| $w, \bar{w}$ | The Wiener process and reverse Wiener process. |
| $\mathbf{f}(\xi, t), g(t)$ | The drift,diffusion coefficient. |
| $\mathbf{f}(\xi, t), g(t)$ | The drift,diffusion coefficient. |
| $\theta$ | The parameters of model $\Phi$. |
| $\epsilon$ | The Gaussian noise $\sim N(0, I)$. |
| $f_{\text{cut}}$ | A cutoff function that ensures $\xi$ remains within the valid range. |
| $\beta_k^e, \mu_k^e, d_k$ | The width, center and threshold parameters. |
| $K$ | The numbers of exponential Gaussian kernels. |
| $f_{at}^e, f_{\mathcal{S}}^e, f_{\mathcal{L}}^e, f_{\Theta}^e, f_{\mathcal{X}}^e$ | Node embedding functions. |
| $f_{cb}^e, f_d^e, f_{dr}^e, f_b^e$ | Edge embedding functions. |
| $l_{at}, l_{\mathcal{S}}, l_{ne}, l_{ee}$ | The length of embedding. |
| $x_i$ | All information of the $i$-th atom |
| $x_i^e, x_{ij}^e$ | The node embedding and edge embedding. |
| $d_{ij}$ | The distance between the $i$-th and $j$-th atom. |
| $L$ | Total number of attention layers. |
| $l$ | Layer index, $l = 0, 1, \ldots, L-1$. |
| $q^{(l)}, k^{(l)}, v^{(l)}$ | Query, key, value. |
| FC | The fully connected layer. |
| $m^{(l)}$ | Aggregated information from other nodes. |
| $w^{(l)}$ | The attention weight. |
| $f_{ln}$ | Adaptive scale function. |
| $f_s$ | Shift function. |
| $\text{MAE}_{\mathcal{L}}, \text{MAE}_{\Theta}, \text{MAE}_{\mathcal{X}}$ | Mean absolute error of $\mathcal{L}, \Theta, \mathcal{X}$. |

Table 5: Settings for different embedding functions.

| Function | $\mu_{\min}$ | $\mu_{\max}$ | $K$ |
|---|---|---|---|
| $f^e_{\mathcal{L}}$ | 0 | 10 | 8 |
| $f^e_{\Theta}$ | 0 | $\pi$ | 8 |
| $f^e_{\mathcal{X}}$ | $-\pi$ | $\pi$ | 8 |
| $f^e_{\mathrm{dr}}$ | 0 | 5 | 16 |

Table 6: Statistics of datasets.

| Dataset | Villin | WW domain | Protein B | BBL | Homeodomain | Trp-cage | BBA |
|---|---|---|---|---|---|---|---|
| Atoms | 577 | 562 | 737 | 710 | 919 | 272 | 504 |
| Amino Acids | 35 | 35 | 47 | 47 | 52 | 20 | 28 |
| Conformations | 2517 | 235 | 1600 | 1600 | 1600 | 1600 | 1600 |

## B  ADDITIONAL DETAILS IN EXPERIMENTS

### B.1  DATASETS

In (Lindorff-Larsen et al., 2011), twelve small, fast-folding proteins (10–80 residues) were simulated using long, atomistic MD in explicit water with a single modified CHARMM force field on the Anton supercomputer. Simulations near the melting temperature yielded repeated folding–unfolding within $100\mu$s-1ms trajectories, giving several folds and unfolds per protein and $\sim$8 ms total data.

We select seven datasets (details in Table 6). Five (Villin, WW domain, Protein B, BBL, and Homeodomain) are used for training, and the remaining two are reserved for testing and generalization evaluation. Because the test sets comprise entirely different proteins, with no sequence or structural overlap with the training proteins, there is no data leakage.

### B.2  HYPERPARAMETERS

There are several hyperparameters in Pandora. As for the optimizer for training, we choose Adam optimizer with a learning rate of 0.001, betas (0.9,0.999), and no weight decay. We set the length of atom type and amino acid type embedding as 32 (i.e., $l_e = l_{\mathcal{S}} = 32$).

For each geometric feature (bond length, bond angle, dihedral angle), we use an embedding of size $K$ for each type. Specifically, there are four types of bond lengths and four types of bond angles, resulting in a total embedding size of $4K$ for each group. For dihedral angles, although there are four unique types as listed above, the definition of each dihedral angle involves overlapping atoms. For instance, the dihedral angle $\mathcal{X}_1$ is defined on $N_i - C\alpha_i - C_i - N_{i+1}$ for each residue $i$. As a result, an atom such as $N_2$ is involved in two consecutive dihedral angles: first as the terminal atom in $N_1 - C\alpha_1 - C_1 - N_2$, and then as the initial atom in $N_2 - C\alpha_2 - C_2 - N_3$. The same applies to $\mathcal{X}_2$ and $\mathcal{X}_3$, where overlapping atoms also lead to duplicated embedding entries. This overlap means that, although there are only four types of dihedral angles, the total number of unique used for dihedral embeddings is $4 + 3 = 7$, because each internal atom is counted twice due to this sharing. Therefore, the total embedding size for dihedral angles is $7K$. In summary, $l_{ne} = l_e + l_{\mathcal{S}} + 4K + 4K + 7K = 184$

Regarding the length of the edge embedding, it is given by $l_{ee} = 1 + 1 + K + K = 26$, as detailed in Appendix A.3. Meanwhile, we employ $L = 5$ attention layers, through which node and edge representations are iteratively updated with information from each other.

### B.3  BASELINES

We compare several baselines to our method, training each on the same datasets and under the same protocol as Pandora to ensure fairness.

**JODO and JODO-L.** JODO (Huang et al., 2024) is capable of generating molecules with varying numbers of atoms, different bond types, and diverse atomic coordinates. However, in our setting, the design of new proteins is not required, and thus, it is unnecessary to generate varying atom counts or bond structures. Consequently, we modify JODO to focus solely on generating atomic coordinates. During training, we compute the mean squared error (MSE) between the denoised coordinates and the

Table 7: Comparison on $\xi_0$ recovery.

| Method | $\text{MAE}_\mathcal{L} \downarrow$ | $\text{MAE}_\Theta \downarrow$ | $\text{MAE}_\mathcal{X} \downarrow$ |
|---|---|---|---|
| JODO | $0.176_{(\pm 0.023)}$ | $0.293_{(\pm 0.003)}$ | $1.415_{(\pm 0.002)}$ |
| JODO-L | $0.093_{(\pm 0.001)}$ | $0.218_{(\pm 0.006)}$ | $1.369_{(\pm 0.007)}$ |
| GearNet | $0.422_{(\pm 0.003)}$ | $0.430_{(\pm 0.004)}$ | $0.753_{(\pm 0.000)}$ |
| STR2STR | $0.526_{(\pm 0.074)}$ | $0.375_{(\pm 0.025)}$ | $1.167_{(\pm 0.084)}$ |
| CONFDIFF | $0.337_{(\pm 0.026)}$ | $0.252_{(\pm 0.002)}$ | $1.083_{(\pm 0.003)}$ |
| CONFROVER | $0.157_{(\pm 0.007)}$ | $0.201_{(\pm 0.006)}$ | $0.702_{(\pm 0.034)}$ |
| Pandora | $\mathbf{0.025}_{(\pm 0.004)}$ | $\mathbf{0.050}_{(\pm 0.000)}$ | $\mathbf{0.374}_{(\pm 0.002)}$ |

Table 8: Ablation study.

| Method | $\text{MAE}_\mathcal{L} \downarrow$ | $\text{MAE}_\Theta \downarrow$ | $\text{MAE}_\mathcal{X} \downarrow$ |
|---|---|---|---|
| - len | $0.168_{(\pm 0.002)}$ | $0.052_{(\pm 0.000)}$ | $0.380_{(\pm 0.002)}$ |
| - angle | $0.026_{(\pm 0.000)}$ | $0.190_{(\pm 0.001)}$ | $0.382_{(\pm 0.001)}$ |
| - torsion | $0.025_{(\pm 0.000)}$ | $0.051_{(\pm 0.000)}$ | $0.688_{(\pm 0.001)}$ |
| - type | $0.025_{(\pm 0.000)}$ | $0.051_{(\pm 0.000)}$ | $0.446_{(\pm 0.009)}$ |

ground truth coordinates to optimize the model. For JODO-L, we further compute the mean squared error between the denoised $\xi_0$ and the true $\xi_0$, i.e., $\xi_0$ loss, enabling JODO-L to more accurately recover the original $\xi_0$.

**GearNet.** We employ the architecture of GearNet to compute node and edge representations, and subsequently predict $\xi_0$ following the approach described in (Zhang et al., 2022).

**STR2STR / CONFDIFF / CONFROVER.** These methods employ SE(3)-based, residue-level models that generate equivariant rotations and translations to reconstruct protein backbones. For CONFROVER, we use its time-independent sampling for fair comparison.

### B.4 $\xi_0$ RECOVERY EXPERIMENT DETAILS

In this section, we present detailed results for each individual dataset. The results in Table 9 closely align with those in Table 1. Overall, Pandora consistently outperforms baseline methods in recovering the lengths and angles of various conformations with minimal fluctuation. This remarkable performance highlights Pandora's ability to effectively capture structural variations and accurately reconstruct conformations.

Among the other methods, JODO-L still consistently outperforms JODO across all three metrics and all five datasets, largely due to the inclusion of the $\xi_0$ loss in JODO-L. Although this enhancement improves its overall performance, its dihedral recovery still shows a substantially higher MAE compared to Pandora. Meanwhile, GearNet achieves a relatively lower $\text{MAE}_\mathcal{X}$, but its performance is hindered by higher MAE values in bond length and bond angle recovery. For the remaining methods, STR2STR shows middling performance with noticeably higher angle errors than Pandora. CONFDIFF narrows the gap somewhat, but still has larger MAEs in both lengths and angles. CONFROVER occasionally competes on angles, yet its bond-length errors remain higher overall.

### B.5 DETAILS OF INFERENCE EXPERIMENTS

We report the Wasserstein distance comparison of all methods across all training datasets in Table 10. The Wasserstein distances between the true $\xi_0$ and the generated $\xi_0$ distributions for each individual dataset are presented in Table 11. The results remain consistent with those in Table 2 and Table 10. Pandora achieves notably low Wasserstein distances, highlighting its exceptional ability to generate realistic native and non-native protein conformations. Meanwhile, JODO and JODO-L do not exhibit a clear superiority over each other across the datasets. As for GearNet, it continues to exhibit high Wasserstein distances in $\mathcal{L}$ and $\Theta$. Additionally, as the number of amino acids increases, GearNet's performance in $\mathcal{L}$ worsens, indicating its inability to effectively capture bond length information, particularly in larger protein conformations. For the remaining methods, STR2STR attains moderate distances but remains above Pandora on most coordinates. CONFDIFF reduces mismatch on a few coordinates yet still trails in both bond-length and angle components. CONFROVER shows occasional competitiveness on isolated terms (e.g., $\mathcal{X}_2$) but generally exhibits larger distances, leading to less faithful distributional recovery. Figure 5 to Figure 39 present the histograms of all datasets for each method.

Table 9: Comparison on $\xi_0$ recovery on inference set.

| Datasets | Villin | WW domain | Protein B | BBL | Homeodomain |
|---|---|---|---|---|---|
| | $\mathrm{MAE}_{\mathcal{L}}$ | | | | |
| JODO | 0.137 (±0.012) | 0.246 (±0.033) | 0.245 (±0.051) | 0.177 (±0.023) | 0.153 (±0.013) |
| JODO-L | 0.078 (±0.001) | 0.141 (±0.019) | 0.129 (±0.013) | 0.083 (±0.006) | 0.080 (±0.007) |
| GearNet | 0.425 (±0.004) | 0.427 (±0.009) | 0.422 (±0.005) | 0.424 (±0.007) | 0.420 (±0.004) |
| STR2STR | 0.472 (±0.064) | 0.466 (±0.060) | 0.584 (±0.070) | 0.564 (±0.106) | 0.512 (±0.069) |
| CONFDIFF | 0.317 (±0.028) | 0.316 (±0.020) | 0.370 (±0.030) | 0.336 (±0.025) | 0.338 (±0.028) |
| CONFROVER | 0.145 (±0.004) | 0.149 (±0.005) | 0.170 (±0.007) | 0.159 (±0.009) | 0.154 (±0.006) |
| Pandora | **0.026** (±0.000) | **0.026** (±0.000) | **0.025** (±0.000) | **0.024** (±0.000) | **0.026** (±0.000) |
| | $\mathrm{MAE}_{\Theta}$ | | | | |
| JODO | 0.280 (±0.004) | 0.351 (±0.007) | 0.335 (±0.002) | 0.290 (±0.003) | 0.266 (±0.006) |
| JODO-L | 0.209 (±0.004) | 0.282 (±0.016) | 0.275 (±0.010) | 0.206 (±0.006) | 0.189 (±0.006) |
| GearNet | 0.431 (±0.003) | 0.432 (±0.005) | 0.429 (±0.006) | 0.430 (±0.007) | 0.427 (±0.004) |
| STR2STR | 0.374 (±0.024) | 0.365 (±0.018) | 0.370 (±0.028) | 0.384 (±0.015) | 0.373 (±0.033) |
| CONFDIFF | 0.250 (±0.002) | 0.228 (±0.006) | 0.246 (±0.003) | 0.244 (±0.000) | 0.250 (±0.002) |
| CONFROVER | 0.190 (±0.002) | 0.196 (±0.003) | 0.211 (±0.005) | 0.203 (±0.006) | 0.198 (±0.005) |
| Pandora | **0.050** (±0.000) | **0.052** (±0.000) | **0.050** (±0.000) | **0.047** (±0.000) | **0.050** (±0.000) |
| | $\mathrm{MAE}_{\mathcal{X}}$ | | | | |
| JODO | 1.349 (±0.013) | 1.700 (±0.032) | 1.593 (±0.009) | 1.400 (±0.007) | 1.309 (±0.017) |
| JODO-L | 1.324 (±0.010) | 1.534 (±0.034) | 1.518 (±0.021) | 1.340 (±0.007) | 1.293 (±0.013) |
| GearNet | 0.738 (±0.008) | 0.821 (±0.020) | 0.805 (±0.011) | 0.758 (±0.012) | 0.711 (±0.005) |
| STR2STR | 1.149 (±0.079) | 1.232 (±0.075) | 1.179 (±0.103) | 1.203 (±0.051) | 1.137 (±0.111) |
| CONFDIFF | 1.068 (±0.005) | 1.150 (±0.003) | 1.110 (±0.004) | 1.088 (±0.002) | 1.060 (±0.002) |
| CONFROVER | 0.639 (±0.020) | 0.741 (±0.024) | 0.775 (±0.034) | 0.710 (±0.038) | 0.669 (±0.028) |
| Pandora | **0.357** (±0.005) | **0.409** (±0.017) | **0.452** (±0.005) | **0.359** (±0.002) | **0.332** (±0.011) |

Table 10: Wasserstein distance between true and generated distribution on training dataset.

| Methods | $\mathcal{L}_1 \downarrow$ | $\mathcal{L}_2 \downarrow$ | $\mathcal{L}_3 \downarrow$ | $\mathcal{L}_4 \downarrow$ | $\Theta_1 \downarrow$ | $\Theta_2 \downarrow$ | $\Theta_3 \downarrow$ | $\Theta_4 \downarrow$ | $\mathcal{X}_1 \downarrow$ | $\mathcal{X}_2 \downarrow$ | $\mathcal{X}_3 \downarrow$ | $\mathcal{X}_4 \downarrow$ |
|---|---|---|---|---|---|---|---|---|---|---|---|---|
| JODO | 0.072 | 0.065 | 0.052 | 0.091 | 0.191 | 0.102 | 0.133 | 0.094 | 0.742 | 0.394 | 1.311 | 0.884 |
| JODO-L | 0.078 | 0.057 | 0.061 | 0.078 | 0.208 | 0.180 | 0.185 | 0.159 | 0.751 | 0.567 | 1.317 | 0.885 |
| GearNet | 1.438 | 1.425 | 1.444 | 1.438 | 1.097 | 1.006 | 0.905 | 0.951 | 2.878 | 2.543 | 1.091 | 2.977 |
| STR2STR | - | - | 0.027 | - | - | 0.047 | 0.054 | - | 0.831 | 0.675 | 1.424 | - |
| CONFDIFF | - | - | 0.025 | - | - | 0.042 | 0.048 | - | 1.073 | 1.160 | 2.744 | - |
| CONFROVER | - | - | 0.023 | - | - | 0.038 | 0.044 | - | 0.234 | **0.093** | 0.218 | - |
| Pandora | **0.023** | **0.020** | **0.017** | **0.022** | **0.022** | **0.021** | **0.018** | **0.014** | **0.121** | 0.153 | **0.088** | **0.163** |

## B.6 DETAILS OF GENERALIZATION EXPERIMENTS

Histograms for dataset Trp-cage is shown in Figure 40.

## B.7 PROBABILITY DENSITY MAPS

Figures 41 and 46 show probability density maps for additional datasets. As illustrated, Pandora's generated densities closely match the ground-truth distributions, demonstrating its strong ability to produce samples that follow the true distribution even for proteins unseen during training.

## C LLM USAGE

We used a Large Language Model (LLM) solely to aid and polish the writing of this paper. The LLM was applied to improve grammar, clarity, and fluency, suggest alternative phrasings for readability.

The LLM did not contribute to research ideation, experimental design, data processing or analysis, model development, or interpretation of results. All technical content, claims, and conclusions were conceived, verified, and finalized by the authors, who reviewed and approved all LLM-edited text to ensure accuracy and faithfulness to the original intent.

Table 11: Wasserstein distance between true and generated distribution on each training dataset.

| Methods | $\mathcal{L}_1 \downarrow$ | $\mathcal{L}_2 \downarrow$ | $\mathcal{L}_3 \downarrow$ | $\mathcal{L}_4 \downarrow$ | $\Theta_1 \downarrow$ | $\Theta_2 \downarrow$ | $\Theta_3 \downarrow$ | $\Theta_4 \downarrow$ | $\mathcal{X}_1 \downarrow$ | $\mathcal{X}_2 \downarrow$ | $\mathcal{X}_3 \downarrow$ | $\mathcal{X}_4 \downarrow$ |
|---|---|---|---|---|---|---|---|---|---|---|---|---|
| | | | | | | Villin | | | | | | |
| JODO | 0.054 | 0.034 | 0.042 | 0.052 | 0.154 | 0.138 | 0.144 | 0.129 | 0.505 | 0.578 | 1.278 | 1.132 |
| JODO-L | 0.055 | 0.046 | 0.043 | 0.071 | 0.146 | 0.079 | 0.114 | 0.080 | 0.494 | 0.491 | 1.262 | 1.115 |
| GearNet | 1.199 | 1.183 | 1.203 | 1.197 | 1.176 | 1.083 | 0.983 | 1.031 | 2.111 | 2.293 | 0.993 | 2.969 |
| STR2STR | - | - | 0.027 | - | - | 0.048 | 0.055 | - | 0.646 | 0.762 | 1.442 | - |
| CONFDIFF | - | - | 0.027 | - | - | 0.050 | 0.057 | - | 0.824 | 0.529 | 2.592 | - |
| CONFROVER | - | - | 0.023 | - | - | 0.039 | 0.045 | - | 0.176 | **0.111** | 0.205 | - |
| Pandora | **0.022** | **0.020** | **0.017** | **0.021** | **0.022** | **0.023** | **0.021** | **0.014** | **0.185** | 0.223 | **0.088** | **0.269** |
| | | | | | | WW domain | | | | | | |
| JODO | 0.092 | 0.069 | 0.066 | 0.098 | 0.243 | 0.215 | 0.214 | 0.179 | 1.377 | 0.621 | 1.309 | 0.592 |
| JODO-L | 0.094 | 0.088 | 0.066 | 0.117 | 0.251 | 0.135 | 0.163 | 0.111 | 1.370 | 0.388 | 1.322 | 0.591 |
| GearNet | 1.200 | 1.184 | 1.204 | 1.198 | 1.188 | 1.083 | 0.983 | 1.033 | 3.234 | 2.096 | 0.787 | 1.731 |
| STR2STR | - | - | 0.028 | - | - | 0.052 | 0.053 | - | 1.347 | 0.747 | 1.310 | - |
| CONFDIFF | - | - | 0.024 | - | - | 0.045 | 0.048 | - | 1.544 | 0.431 | 1.936 | - |
| CONFROVER | - | - | 0.024 | - | - | 0.042 | 0.044 | - | 0.388 | 0.121 | 0.165 | - |
| Pandora | **0.024** | **0.021** | **0.018** | **0.024** | **0.019** | **0.015** | **0.014** | **0.013** | **0.139** | **0.114** | **0.109** | **0.073** |
| | | | | | | Protein B | | | | | | |
| JODO | 0.102 | 0.079 | 0.088 | 0.115 | 0.248 | 0.210 | 0.218 | 0.178 | 0.784 | 0.536 | 1.414 | 0.548 |
| JODO-L | 0.088 | 0.082 | 0.063 | 0.123 | 0.226 | 0.120 | 0.156 | 0.106 | 0.788 | 0.284 | 1.410 | 0.555 |
| GearNet | 1.593 | 1.583 | 1.601 | 1.595 | 1.044 | 0.957 | 0.853 | 0.895 | 3.299 | 2.624 | 1.135 | 3.008 |
| STR2STR | - | - | 0.026 | - | - | 0.045 | 0.054 | - | 0.801 | 0.414 | 1.480 | - |
| CONFDIFF | - | - | 0.025 | - | - | 0.041 | 0.045 | - | 1.119 | 2.371 | 3.436 | - |
| CONFROVER | - | - | 0.022 | - | - | 0.037 | 0.045 | - | 0.228 | 0.093 | 0.268 | - |
| Pandora | **0.022** | **0.020** | **0.016** | **0.022** | **0.024** | **0.022** | **0.017** | **0.014** | **0.172** | **0.080** | **0.120** | **0.297** |
| | | | | | | BBL | | | | | | |
| JODO | 0.079 | 0.055 | 0.055 | 0.068 | 0.210 | 0.179 | 0.180 | 0.160 | 0.582 | 0.493 | 1.249 | 0.796 |
| JODO-L | 0.064 | 0.055 | 0.045 | 0.069 | 0.177 | 0.090 | 0.116 | 0.086 | 0.570 | 0.307 | 1.241 | 0.806 |
| GearNet | 1.593 | 1.583 | 1.601 | 1.595 | 1.037 | 0.953 | 0.856 | 0.898 | 3.054 | 2.735 | 1.319 | 3.303 |
| STR2STR | - | - | 0.025 | - | - | 0.044 | 0.050 | - | 0.637 | 0.610 | 1.393 | - |
| CONFDIFF | - | - | 0.023 | - | - | 0.034 | 0.044 | - | 0.947 | 1.064 | 2.784 | - |
| CONFROVER | - | - | 0.021 | - | - | 0.036 | 0.041 | - | 0.200 | **0.089** | 0.239 | - |
| Pandora | **0.023** | **0.021** | **0.018** | **0.022** | **0.022** | **0.021** | **0.019** | **0.014** | **0.050** | 0.127 | **0.065** | **0.106** |
| | | | | | | Homeodomain | | | | | | |
| JODO | 0.065 | 0.047 | 0.052 | 0.055 | 0.185 | 0.159 | 0.168 | 0.147 | 0.508 | 0.607 | 1.333 | 1.356 |
| JODO-L | 0.060 | 0.054 | 0.042 | 0.073 | 0.156 | 0.087 | 0.118 | 0.086 | 0.489 | 0.499 | 1.319 | 1.351 |
| GearNet | 1.604 | 1.591 | 1.610 | 1.604 | 1.041 | 0.953 | 0.852 | 0.897 | 2.694 | 2.967 | 1.221 | 3.872 |
| STR2STR | - | - | 0.027 | - | - | 0.046 | 0.055 | - | 0.723 | 0.840 | 1.495 | - |
| CONFDIFF | - | - | 0.028 | - | - | 0.042 | 0.048 | - | 0.929 | 1.405 | 2.971 | - |
| CONFROVER | - | - | 0.023 | - | - | 0.036 | 0.044 | - | 0.181 | **0.051** | 0.212 | - |
| Pandora | **0.023** | **0.020** | **0.016** | **0.021** | **0.021** | **0.022** | **0.020** | **0.016** | **0.059** | 0.220 | **0.055** | **0.072** |

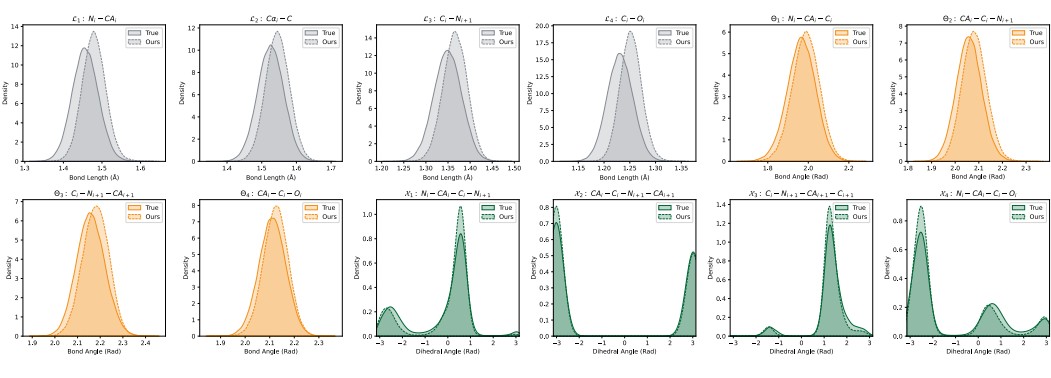

Figure 5: Pandora-generated histograms for dataset Villin.

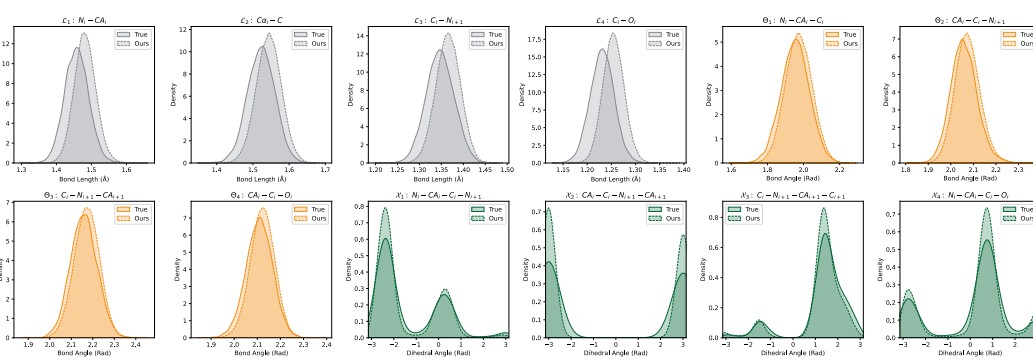

Figure 6: Pandora-generated histograms for dataset WW domain.

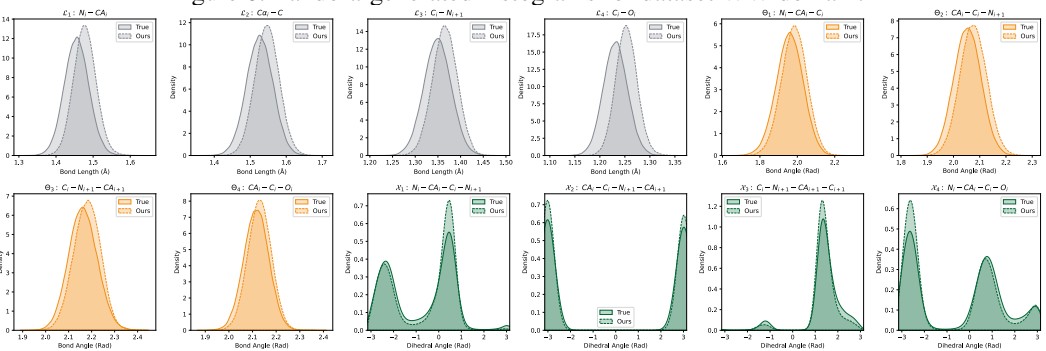

Figure 7: Pandora-generated histograms for dataset Protein B.

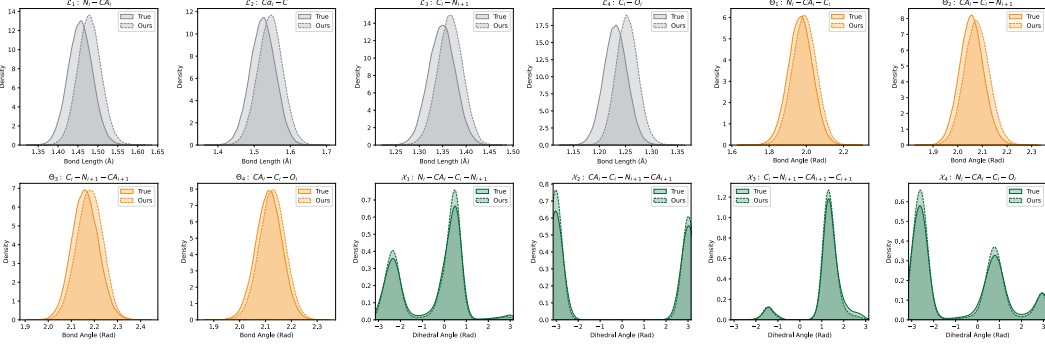

Figure 8: Pandora-generated histograms for dataset BBL.

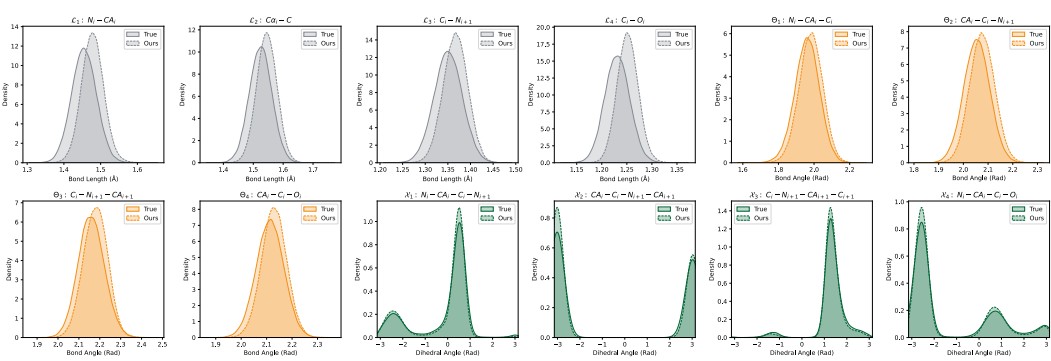

Figure 9: Pandora-generated histograms for dataset Homeodomain.

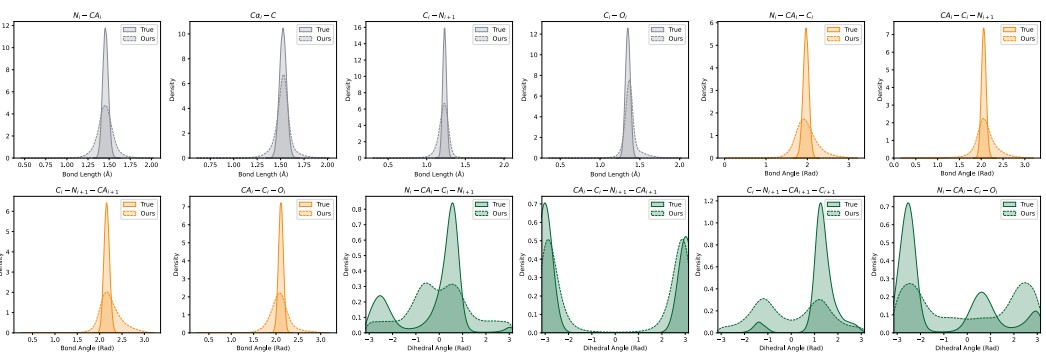

Figure 10: JODO-generated histograms for dataset Villin.

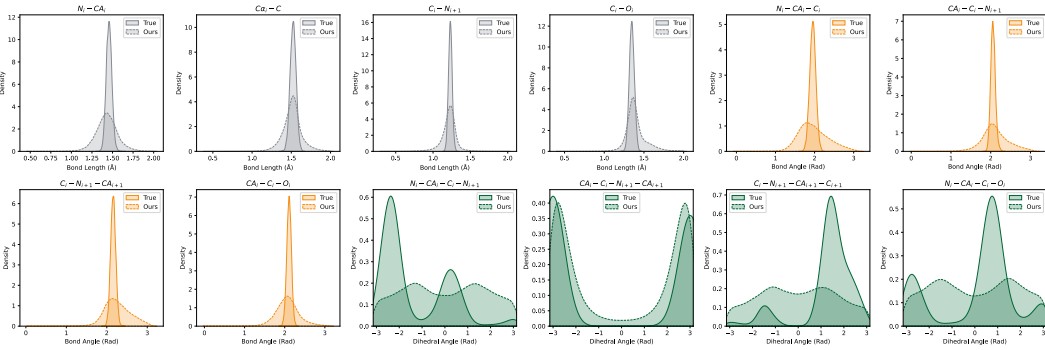

Figure 11: JODO-generated histograms for dataset WW domain.

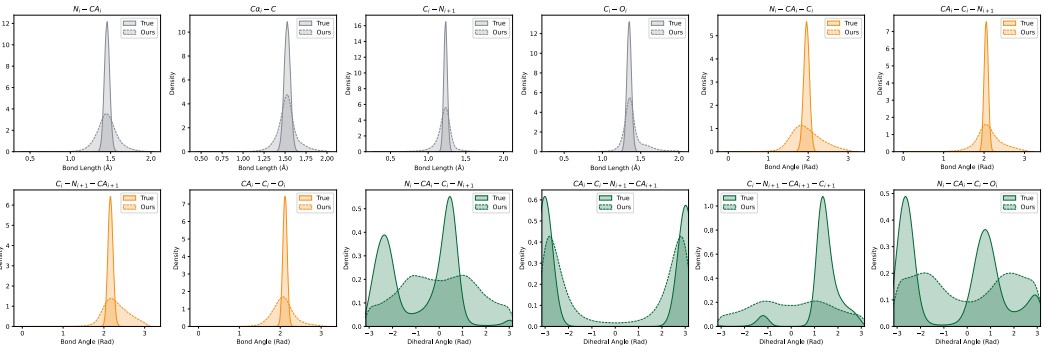

Figure 12: JODO-generated histograms for dataset Protein B.

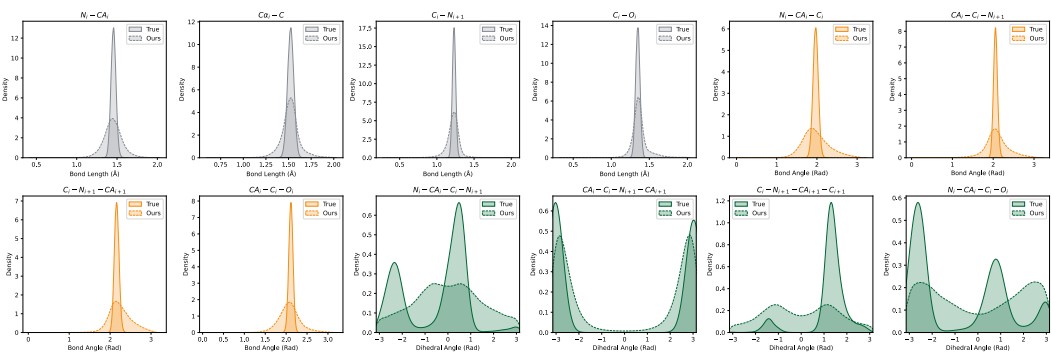

Figure 13: JODO-generated histograms for dataset BBL.

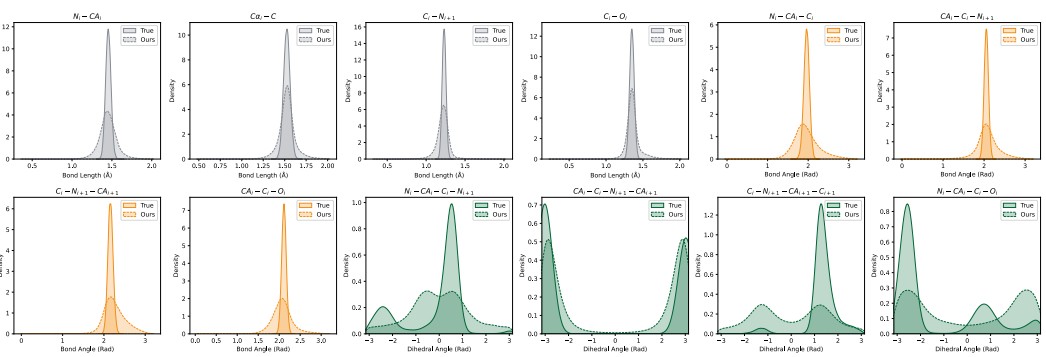

Figure 14: JODO-generated histograms for dataset Homeodomain.

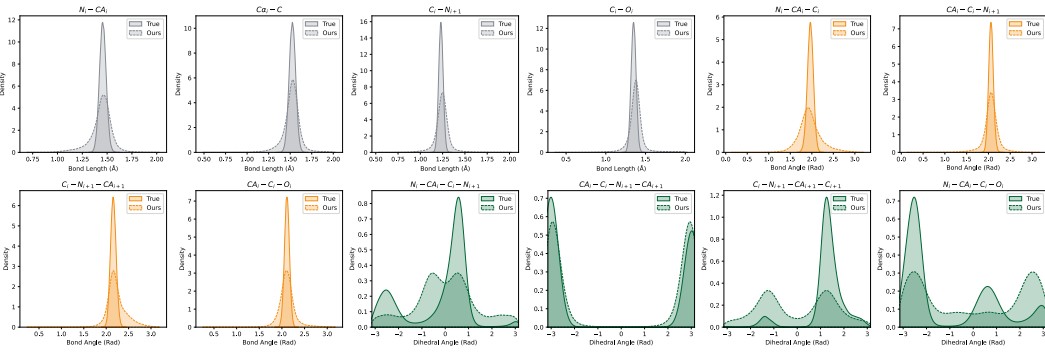

Figure 15: JODO-L-generated histograms for dataset Villin.

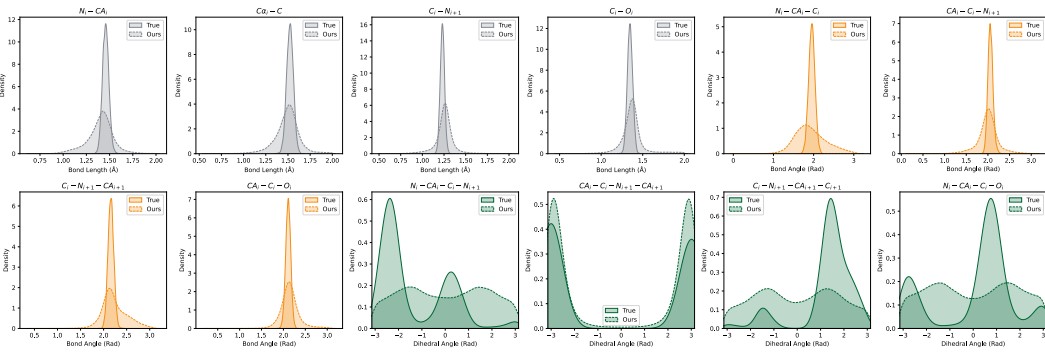

Figure 16: JODO-L-generated histograms for dataset WW domain.

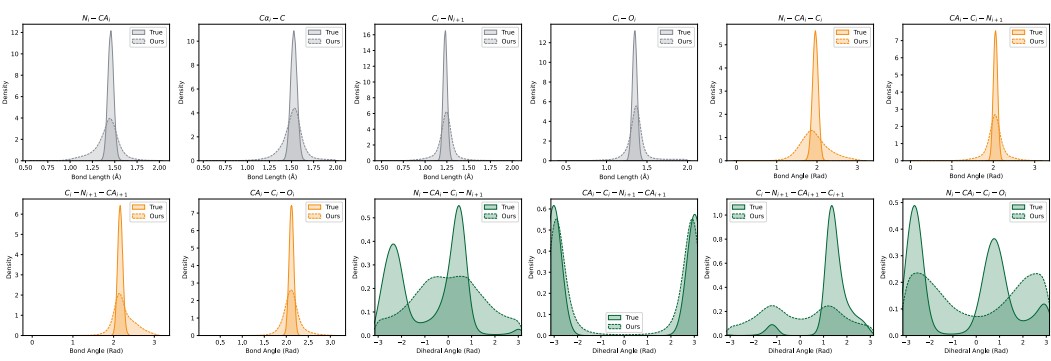

Figure 17: JODO-L-generated histograms for dataset Protein B.

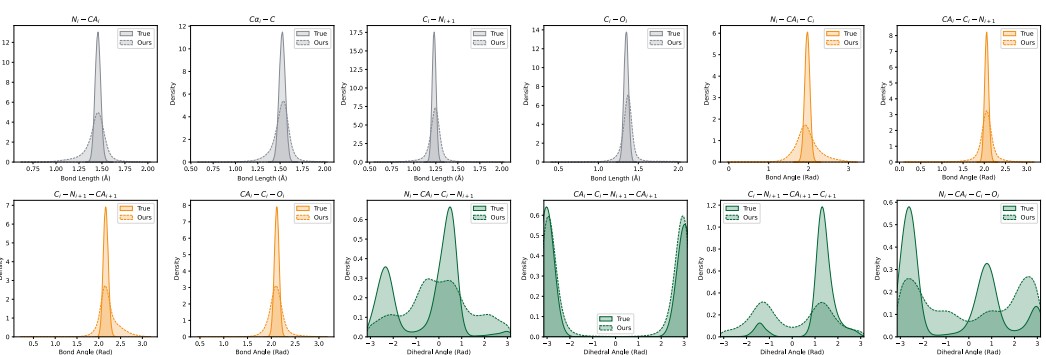

Figure 18: JODO-L-generated histograms for dataset BBL.

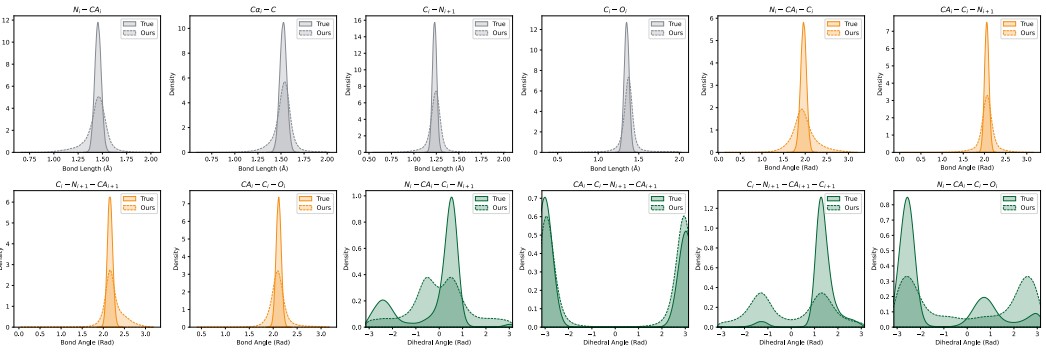

Figure 19: JODO-L-generated histograms for dataset Homeodomain.

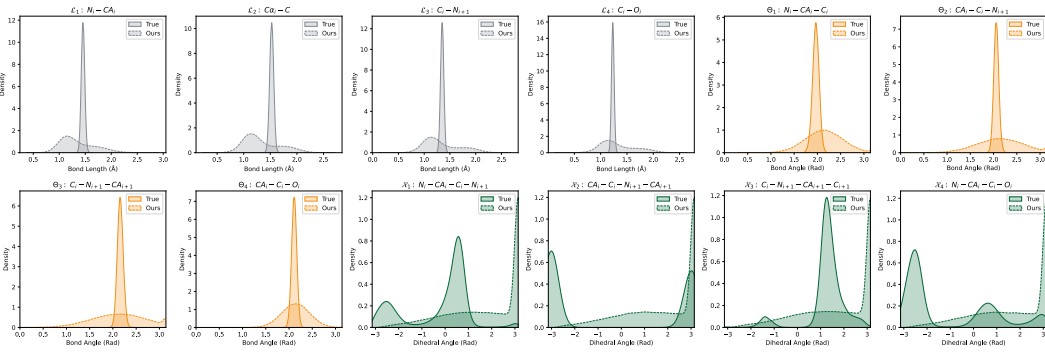

Figure 20: GearNet-generated histograms for dataset Villin.

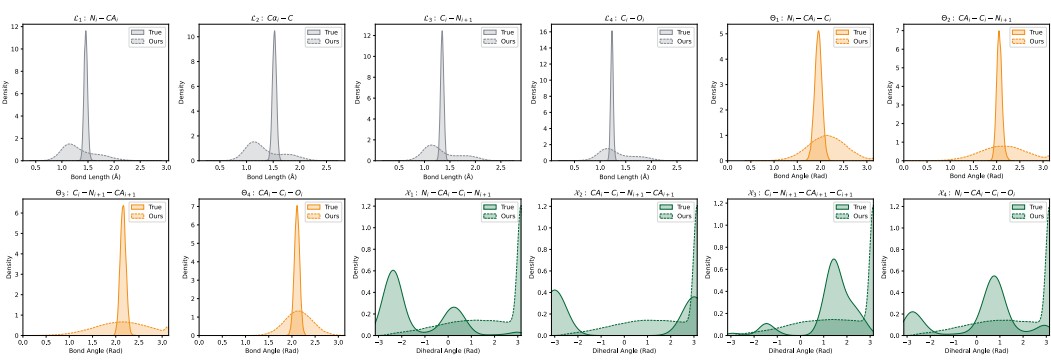

Figure 21: GearNet-generated histograms for dataset WW domain.

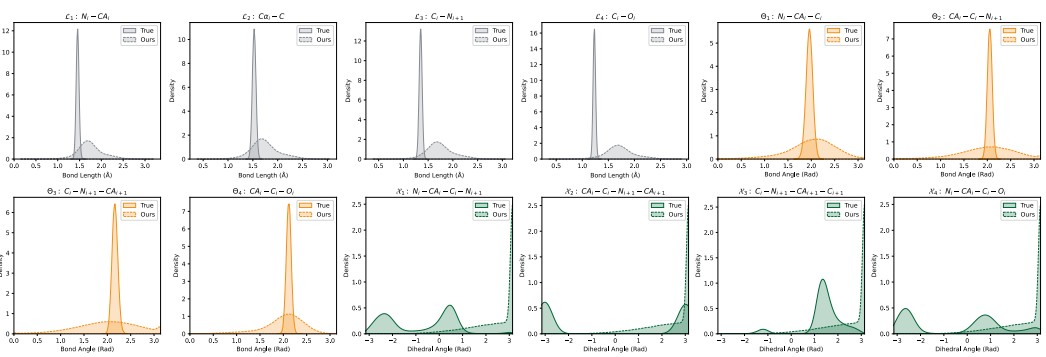

Figure 22: GearNet-generated histograms for dataset Protein B.

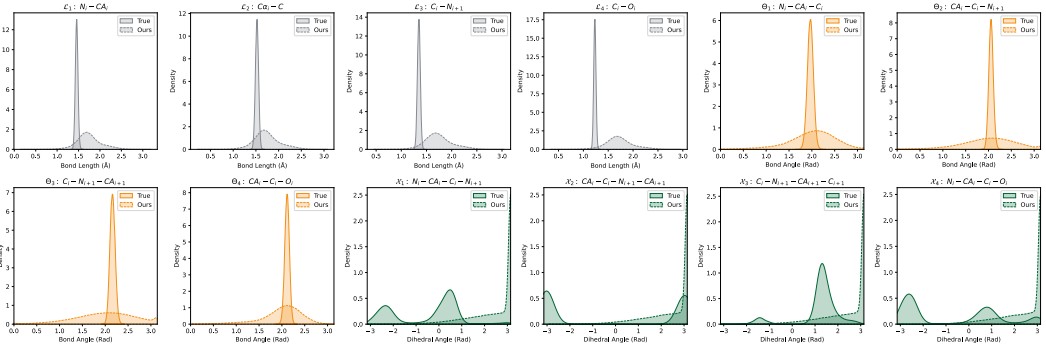

Figure 23: GearNet-generated histograms for dataset BBL.

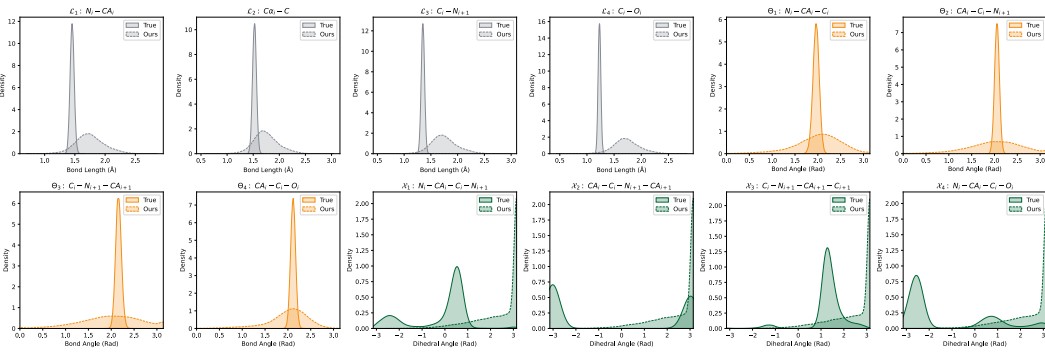

Figure 24: GearNet-generated histograms for dataset Homeodomain.

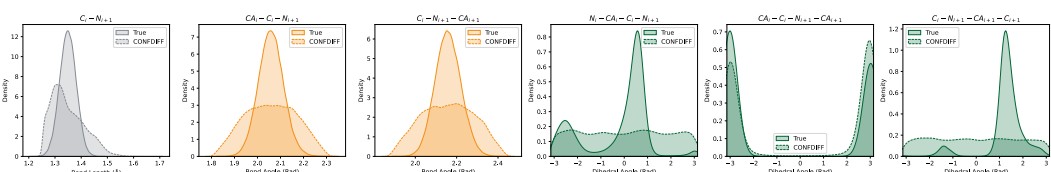

Figure 25: STR2STR-generated histograms for dataset Villin.

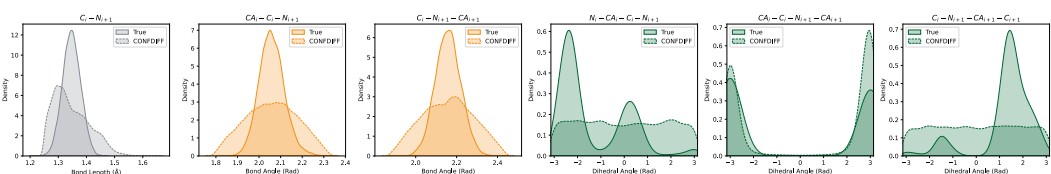

Figure 26: STR2STR-generated histograms for dataset WW domain.

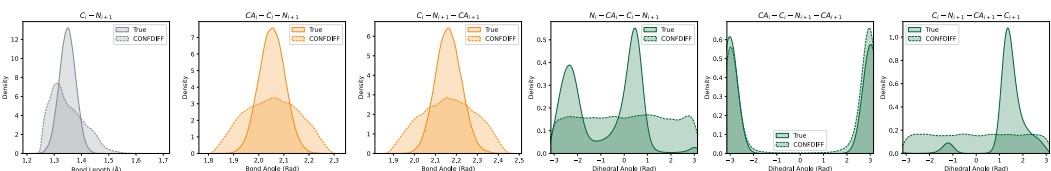

Figure 27: STR2STR-generated histograms for dataset Protein B.

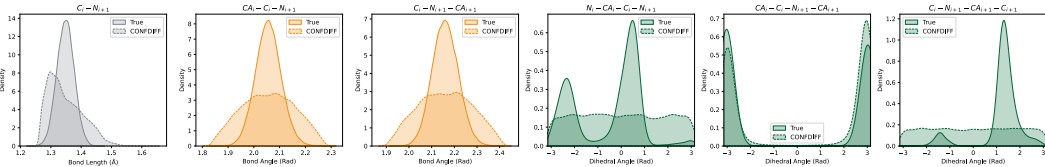

Figure 28: STR2STR-generated histograms for dataset BBL.

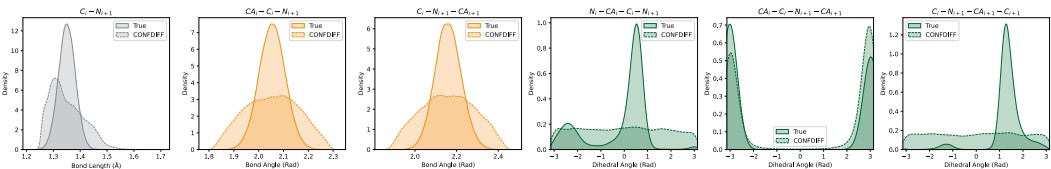

Figure 29: STR2STR-generated histograms for dataset Homeodomain.

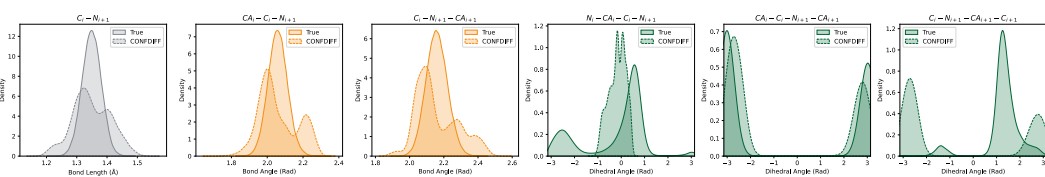

Figure 30: CONFDIFF-generated histograms for dataset Villin.

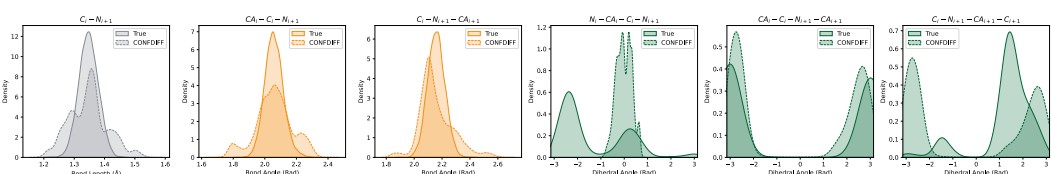

Figure 31: CONFDIFF-generated histograms for dataset WW domain.

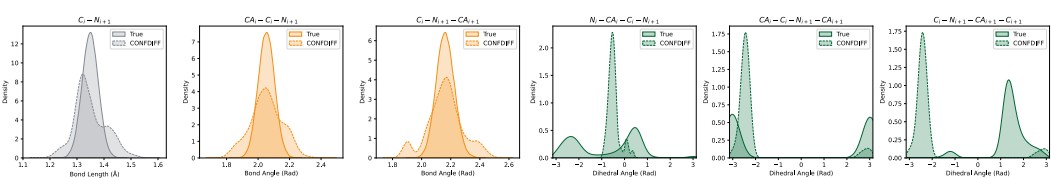

Figure 32: CONFDIFF-generated histograms for dataset Protein B.

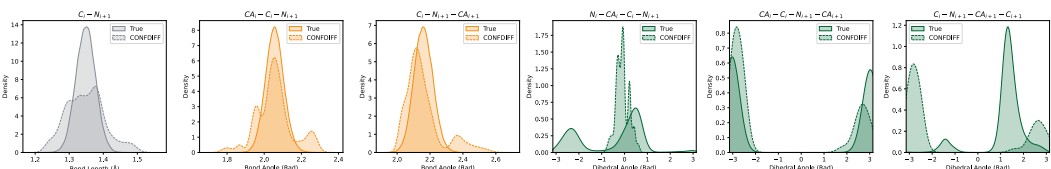

Figure 33: CONFDIFF-generated histograms for dataset BBL.

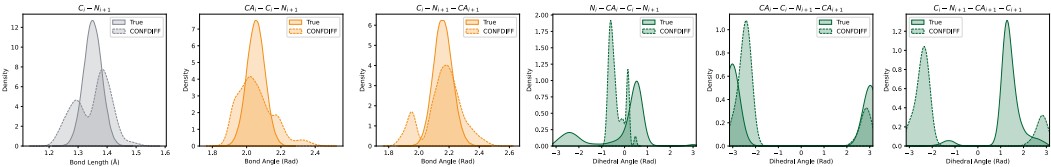

Figure 34: CONFDIFF-generated histograms for dataset Homeodomain.

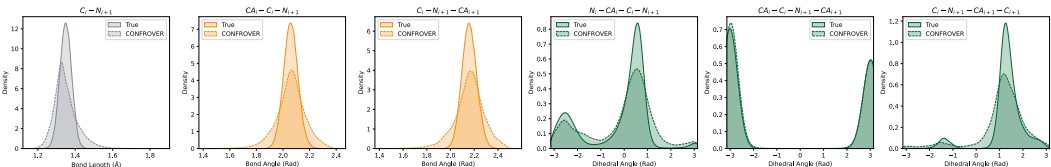

Figure 35: CONFROVER-generated histograms for dataset Villin.

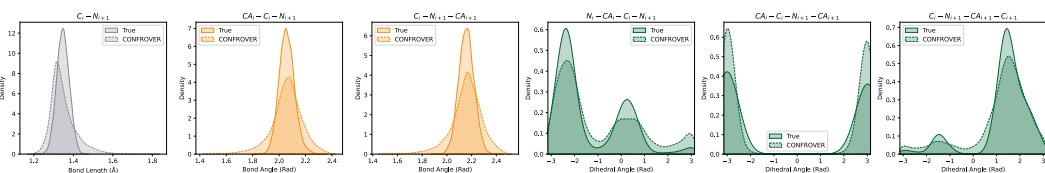

Figure 36: CONFROVER-generated histograms for dataset WW domain.

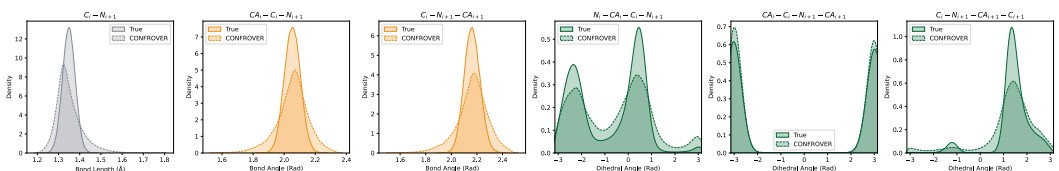

Figure 37: CONFROVER-generated histograms for dataset Protein B.

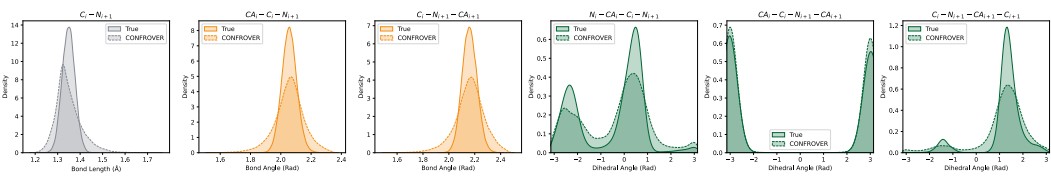

Figure 38: CONFROVER-generated histograms for dataset BBL.

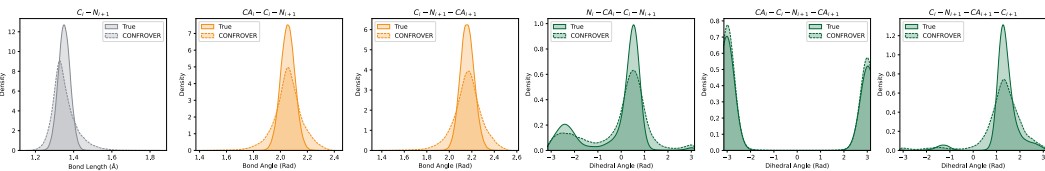

Figure 39: CONFROVER-generated histograms for dataset Homeodomain.

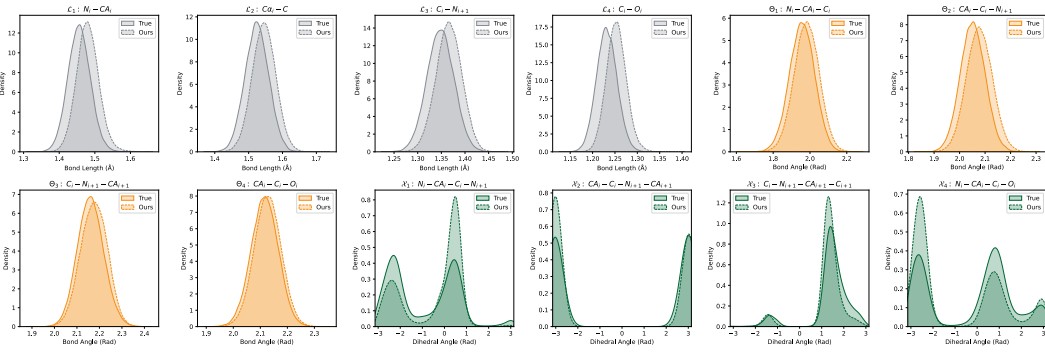

Figure 40: Pandora-generated histograms for dataset Trp-cage.

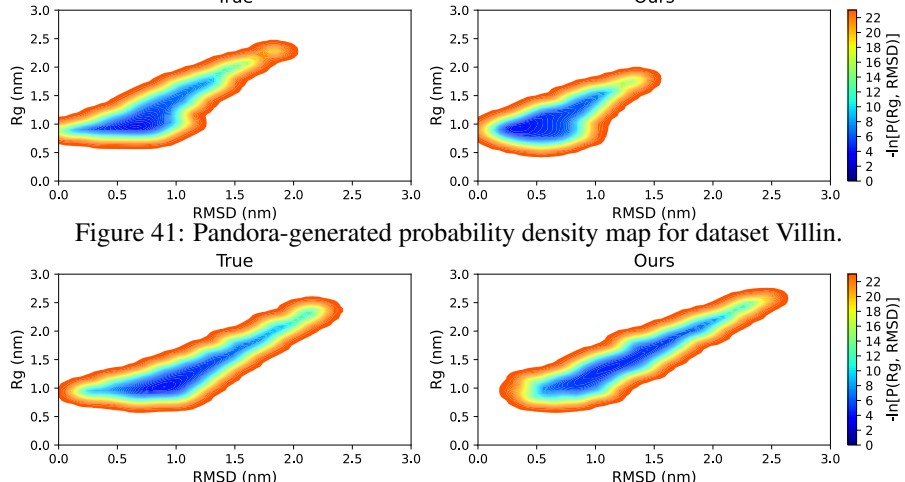

Figure 41: Pandora-generated probability density map for dataset Villin.

Figure 42: Pandora-generated probability density map for dataset WW domain.

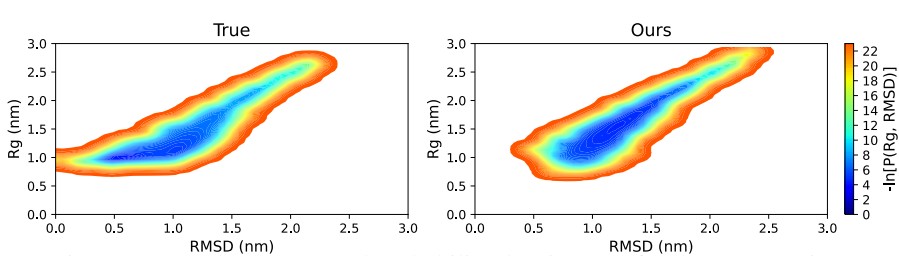

Figure 43: Pandora-generated probability density map for dataset Protein B.

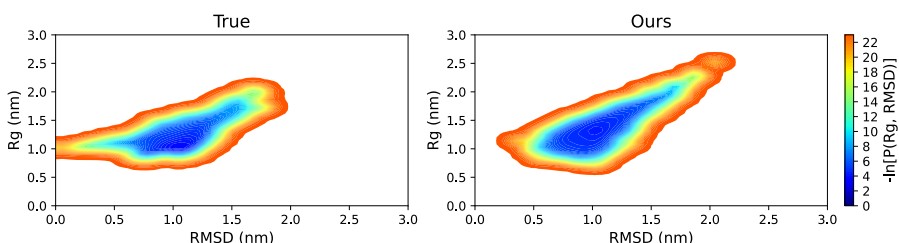

Figure 44: Pandora-generated probability density map for dataset BBL.

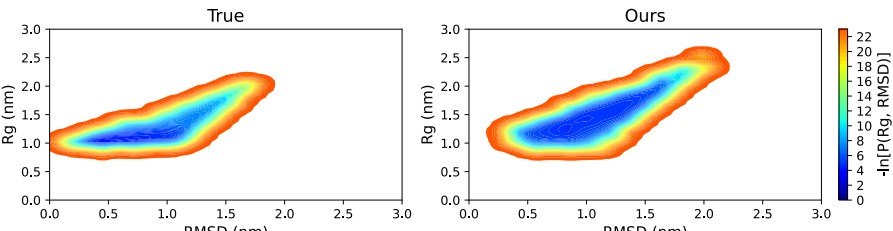

Figure 45: Pandora-generated probability density map for Homeodomain.

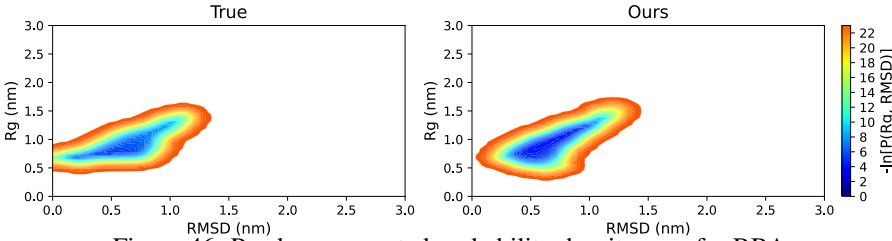

Figure 46: Pandora-generated probability density map for BBA.

