# OpenReview forum: "PANDORA: Diffusion-based Protein Conformation Generation"
_ICLR.cc/2026/Conference — Submitted to ICLR 2026_

### Official Review · Reviewer_2f2J · 2025-10-16

**Soundness:** 2
**Presentation:** 4
**Contribution:** 2
**Rating:** 4
**Confidence:** 4

**Summary:**

This paper introduces Pandora, a diffusion-based generative framework that operates in protein internal coordinates—bond lengths, bond angles, and dihedral angles—to produce diverse protein conformations. Experiments across multiple MD datasets, with comparisons against several baselines, show that Pandora achieves superior conformation reconstruction and distributional fidelity.

**Strengths:**

1. The paper is well organized and clearly stated, especially Section 1 & 2.
2. By modeling in angular space, Pandora preserves structural accuracy while enabling more flexible architectures, removing the constraint of equivariance.

**Weaknesses:**

1. Using internal coordinates for molecular conformation generation is highly efficient for small molecules, but it poses challenges for large proteins:
  a). The dimensionality of internal coordinates grows rapidly and can far exceed that of Cartesian coordinates in large proteins.
  b). Errors accumulate during coordinate transforms—when reconstructing Cartesian coordinates from internal angles, small angular errors propagate along the chain, resulting in large deviations in terminal atoms.
  c). Internal coordinates capture only local geometry, so constructing residues that are distant along the sequence but spatially adjacent is likely to cause steric clashes.
Do authors consider these problems?
2. It appears the authors directly cutoff variables to suitable range, which may introduce discontinuities and unstable boundary derivatives. Would a diffusion defined on an appropriate Riemannian manifold be a more suitable choice? Have the authors considered this alternative?
3. As the paper notes, the baselines operate at the residue level, implicitly or explicitly fixing subsets of bond lengths and bond angles. By contrast, Pandora is an atomic-level model. Consequently, many of the baseline comparison metrics are not directly meaningful. Could the authors report Pandora’s structure-quality metrics —e.g., TM-score, RMSD, GDT-TS, IDDT, Cα clash rate, and peptide-bond break frequency—to provide a overall, assessment?

**Questions:**

The same as weaknesses. If the authors can satisfactorily address the concerns above, I would be inclined to increase my score.

---

### Official Review · Reviewer_XmKN · 2025-11-01

**Soundness:** 3
**Presentation:** 3
**Contribution:** 2
**Rating:** 2
**Confidence:** 5

**Summary:**

The paper have strong motivation. The main challenges of this research is sampling diverse conformations, not a single point estimation, They proposed diffusion with noise injection across reverse steps to solve this challenge. The authors did MAE and Wasserstein distance to evaluate their method. However, no energy-based validity, steric-clash rates or experimental comparisons are reported.

1.Node/edge embeddings use Gaussian smearing with fixed grids. Did you tune K, \mu_min, \mu_max per protein class, and how does performance change with learned (e.g., radial basis) centers vs. fixed ones?
2. Beyond MAE/Wasserstein, did you compute steric clash rates, Ramachandran distributions, or energy (e.g., with a force field) to corroborate “physically plausible”? If not, can you add these checks?
3. You note a small right-shift in L (~0.025 Å). Does this accumulate along chains (e.g., drift in end-to-end distance) or vanish after reconstruction to 3D? Any impact on Rg/RMSD beyond the shown density maps?
4. How robust are results across sequence-diverse and longer proteins (>80 aa), and what happens under domain shifts (membrane proteins, IDPs)?

**Strengths:**

The paper’s strengths are clear problem framing and a technically coherent choice to model stereochemistry in ξ-space (bond lengths/angles/dihedrals) with step-wise regulation, which naturally supports diversity while maintaining plausible geometry. Empirically it delivers solid wins: lower MAEs on L/Θ/X recovery, distributional alignment via Wasserstein distances, and ablations that link Gaussian-smeared geometric features to accuracy; the generalization check on unseen proteins (e.g., Trp-cage, BBA) is a nice touch.

**Weaknesses:**

validation leans on geometry statistics without independent physics/quality checks (clashscore, Ramachandran, energies), so “biophysical plausibility” isn’t fully established. The repeated cutoff/clamping during sampling could bias the stationary distribution but isn’t analyzed; data scale is narrow (small proteins), and head-to-head comparisons against the strongest coordinate-space diffusion baselines with structure-quality metrics are limited. There’s also no runtime/efficiency accounting or downstream utility test after reconstructing 3D (e.g., RMSD/Rg or functional tasks).

**Questions:**

1.Node/edge embeddings use Gaussian smearing with fixed grids. Did you tune K, \mu_min, \mu_max per protein class, and how does performance change with learned (e.g., radial basis) centers vs. fixed ones?
2. Beyond MAE/Wasserstein, did you compute steric clash rates, Ramachandran distributions, or energy (e.g., with a force field) to corroborate “physically plausible”? If not, can you add these checks?
3. You note a small right-shift in L (~0.025 Å). Does this accumulate along chains (e.g., drift in end-to-end distance) or vanish after reconstruction to 3D? Any impact on Rg/RMSD beyond the shown density maps?
4. How robust are results across sequence-diverse and longer proteins (>80 aa), and what happens under domain shifts (membrane proteins, IDPs)?

---

### Official Review · Reviewer_9Nfd · 2025-11-01

**Soundness:** 2
**Presentation:** 2
**Contribution:** 2
**Rating:** 2
**Confidence:** 4

**Summary:**

This paper presents PANDORA, a diffusion-based generative model that operates in the internal coordinate space of protein backbones (bond lengths, bond angles, and dihedral torsions). The goal is to generate protein conformations with more physically realistic local geometry and improved alignment with molecular dynamics (MD) ensemble distributions. Pandora uses a graph-based transformer architecture to encode geometric information at both node and edge levels, and performs diffusion directly in this internal coordinate parameterization. Experiments on fast-folding proteins demonstrate that generating conformations in internal coordinate space leads to more accurate recovery of bond lengths, angles, and backbone torsions, both in terms of prediction error and distributional similarity to MD ensembles. These results suggest that explicitly modeling internal degrees of freedom can enhance local geometric fidelity compared to residue-level coordinate generation.

**Strengths:**

- The paper makes a valid and important observation that many residue-level generative models overlook internal backbone degrees of freedom, which are crucial for accurately modeling protein conformations.
- Modeling in internal coordinates (bond lengths, bond angles, and dihedral torsions) is a reasonable approach to improving local geometric accuracy and preserving physically meaningful conformations.
- Empirical results, evaluated using the authors’ proposed metrics, consistently show that explicitly incorporating internal coordinates leads to measurable improvements in local geometry and conformational ensemble quality.

**Weaknesses:**

1. Although the paper aims to explore non-native conformation generation, specifically for protein folding, it presents limited architectural, modeling, or analytical innovation compared to existing protein conformation generation models.
2. While the method claims atomic-level modeling by diffusing over bond lengths, angles, and dihedral torsions, it only models backbone atoms and does not include side-chain degrees of freedom. This makes the "full atomic freedom" claim less convincing, especially given that models like AlphaFold3 and Boltz explicitly model both backbone and side chains in full degree of freedom.
3. The experimental evaluation is restricted to a subset of existing benchmarks (5 training + 2 test proteins from fast-folding datasets), which has present in Str2Str, BioEmu, ConfDiff, or EquiJump. The author formed their own analysis focuses on recovery MSE and distributional accuracy in internal coordinates, the results are limited and are not strongly demonstrating that Pandora can model folding/unfolding pathways, generalize to new proteins, or handle larger systems.
4. Clarity: The paper would benefit from substantial revision for clarity. The methodology section is dense and difficult to follow, and common components such as diffusion training and sampling could be presented in a more standard and simplified manner.

**Questions:**

### Model

1. The paper uses a $f_{cut}$ function to clip angle and torsion values within defined ranges. This seems a bit ad-hoc. Have the authors considered defining the diffusion process directly in the angle or torsional space (e.g., https://arxiv.org/abs/2206.01729), where the periodicity is naturally handled?
2. Several Method details are unclear:
    - The paper states that the model "smooths discrete data (bond length, bond angle, dihedral angle) into continuous distributions." Why are these values described as discrete?
    - What is meant by the "distance relation" in line 282?
    - In Equation (6), should there be a softmax or normalization?
    - In line 299, what does the variable *h* represent?

3. Since the baselines (Str2Str, ConfDiff, ConfRover) were re-trained, could the authors provide more details on their reproduction? For example: specific model variant, number of parameters, training epochs, optimizer, learning rate, and compute resources.
    - For Str2Str, what forward noising cutoff was used?
    - For ConfDiff, was classifier-free guidance used as in the original paper?

### Experiments and results

1. $\xi_0$ Recovery Task Definition: Could the authors explain the task in more details? How are samples generated for each model? Which values are used as reference for MAE computation? How many samples are generated per protein?
2. The paper states that a transferable setup is used (five proteins for training, two for testing) in Experimental setup. However, primary results in Sections 4.2–4.5 are on the training split, with only Section 4.5 showing held-out test results. This difference should be more clearly stated for an accurate interpretation of the results.

3. BioEmu (https://www.science.org/doi/10.1126/science.adv9817) is common diffusion-based protein conformation baseline that has used for fast-folding proteins. Boltz (https://www.biorxiv.org/content/10.1101/2025.06.14.659707v1) is another diffusion-based model, while originally used for protein folding, their all atom diffusion architecture can serve another strong baseline for this task.

4. The free energy surfaces (e.g., Figure 4) are shown in low-dimensional projections of RMSE and Rg, where a standard practice for fast folding dataset is to project coordinate onto TICA (time-lagged independent components), which captures the slow collective motions (e.g., folding/unfolding). Could the authors provide results based on TICA coordinates or similar analysis as in BioEmu and EquiJump (https://arxiv.org/abs/2410.09667)?

5. Evaluation is limited to bond/torsion-space metrics where the Pandora’s are specifically optimized, but lacks coordinate-based assessments (e.g., clashes, residue-residue contacts) needed to evaluate full 3D structural validity and ensemble fidelity.

7. In Table 6: where do the reported “number of conformations” come from, and why does the WW domain contain far less  conformations than other proteins?

### Other comments

1. The introduction can be more concise and focused. For example, broad statements such as "Beyond structural prediction… sparking a surge of innovative research… showcase the transformative potential of deep learning in addressing complex biological problems" feel generic and distract from the main topic. In the abstract and introduction, protein design models are mentioned but not clearly discussed in the context of protein structure/conformation generation.

2. Line 229: The notation $\{\xi\}_{t=0}^T$ is typically used to denote a discrete set, which may be inappropriate when referring to a continuous-time variable.

3. Line 304: typo $x_i^e \to x_i$.

4. Table 9: labeled as "inference set", but the listed proteins are the same five used during training.

---

### Official Review · Reviewer_RRQ5 · 2025-11-02

**Soundness:** 3
**Presentation:** 2
**Contribution:** 2
**Rating:** 2
**Confidence:** 3

**Summary:**

This paper proposes Pandora, a diffusion-based model to generate diverse, plausible native and non-native protein conformations, filling gaps in existing methods. It uses a conditional transformer and integrates structural info (bond lengths, angles) for validity. Experiments show it outperforms baselines and generalizes to unseen proteins.

**Strengths:**

- This study investigates the usage of diffusion models to generate the protein conformations and the experiments show its effectiveness.
- The introduction of related knowledge is comprehensive.

**Weaknesses:**

- The novelty of the method is limited. I think the novelty of denoising the bond lenghts/angles with diffusion models is not a very innovative.
- There is a large space for improvement of the writing. I think the authors spend too many words discussing the most basic things.
	- In Section 1.1, I do not think the **motivation** is the `motivation' of this study, since it does not motivate any design of this study. Instead it is discussing why researchers study the problem of conformation generation.
	- In Section 1.1, line 90, does this study solves the problem of folding pathway? Maybe the folding pathway is something totally different from the conformations.
	- In Section 1.2, the authors discuss the **challenges** and **contributions**. The challenge looks more like a basic introduction of the problem (conformation generation), and the contribution is too general to see what is special in this study.
	- In Section 1.2, the first contribution contains some factual error. There have been several methods investigating utilizing AF3 to generate all-atom conformations, (although they may be not very successful), e.g., MSA-subsampling.
	- In Section 3, the manuscript spends several words introduce the basic knowledge of diffusion models, attentino mechanisms, network architectures, and inference algorithms. I think the authors can use more words to discuss what makes this study different from baselines and other methods.

**Questions:**

- I would suggest the authors reduce the first 6 pages to fewer than 4 pages. There is too much basic knowledge which I think we should assume the readers have basic understandings of this area.

---

### Meta-Review · Area_Chair_DauW · 2026-01-07

**Summary:**

The paper proposes PANDORA, a diffusion model operating on internal coordinates (bond lengths, angles, torsions) to generate diverse protein backbone conformations. The initial review status is unanimously negative (scores: 2, 2, 2, 4). While reviewers acknowledged the valid motivation of modeling non-native conformations and the intuitive choice of angular space, there is a strong consensus on critical weaknesses: limited novelty compared to existing molecular diffusion methods, the misleading "all-atom" claim (side-chains are missing), lack of essential physical validity checks (clashes, Ramachandran), and insufficient comparison against state-of-the-art baselines like BioEmu or Boltz.

**Reviewer Concerns:**

The authors did not provide any rebuttal

**Reviewer Scores:**

I predict the reviewer score to remain the same (4,2,2,2) as no rebuttal was provided

---

### Decision · Program_Chairs · 2026-01-26

Reject